# Anemoside B4 Rectal Thermosensitive In Situ Gel to Treat Ulcerative Colitis by Overcoming Oral Bioavailability Barriers with Absorption Enhancer-Assisted Delivery

**DOI:** 10.3390/pharmaceutics17111400

**Published:** 2025-10-29

**Authors:** Xiaomeng Lei, Canjian Wang, Mingyan Xia, Guansheng Zhang, Tangxun Wang, Yang Chen, Yufang Huang, Tiantian Wang, Dongxun Li, Wenliu Zhang, Guosong Zhang

**Affiliations:** 1National Engineering Research Center of Chinese Medicine Solid Preparation Manufacturing Technology, Jiangxi University of Chinese Medicine, Nanchang 330006, China; 2School of Pharmacy, Jiangxi University of Chinese Medicine, Nanchang 330006, China

**Keywords:** thermosensitive in situ gel, anemoside B4, absorption enhancer, rectal delivery, ulcerative colitis, bioavailability

## Abstract

**Background:** Anemoside B4 (AB4), the major bioactive saponin from Pulsatilla chinensis, exhibits anti-inflammatory, anti-tumor, anti-apoptotic, and analgesic properties. However, its clinical translation for ulcerative colitis (UC) is constrained by poor epithelial permeability and low oral bioavailability. **Objective:** This study’s objective was to engineer and optimize thermosensitive rectal in situ gels (ISGs) of AB4, incorporating suitable absorption enhancers to improve mucosal permeation, bioavailability, and therapeutic efficacy against UC. **Methods:** Screening of effective permeation enhancers was conducted using Caco-2 cell monolayers and Franz diffusion cells. Critical formulation variables such as poloxamer 407 (P407), poloxamer 188 (P188), and hydroxypropyl methyl cellulose (HPMC) were optimized, employing single-factor experiments coupled with the Box–Behnken design response surface methodology (BBD-RSM). Comprehensive characterization encompassed in vitro release kinetics, in vivo pharmacokinetics, rectal tissue tolerability, rectal retention time, and pharmacodynamic efficacy in a UC model. **Results:** We used 2.5% hydroxypropyl-β-cyclodextrin (HP-β-CD) and 1.0% sodium caprate (SC) as the appropriate absorption enhancers, and the amounts of P407, P188, and HPMC were 17.41%, 4.07%, and 0.44%, respectively, to yield the corresponding in situ gels HP-β-CD-AB4-ISG and SC-AB4-ISG. The gel characterization, such as gelation temperature, gelation time, pH, gelation strength, etc., was in accordance with requirements. The ISGs did not stimulate or damage rectal tissue and remained in the rectum for a prolonged period. More importantly, an improvement in bioavailability and alleviation of UC were noted. **Conclusion:** Absorption enhancer-assisted, poloxamer-based thermosensitive rectal ISGs provide a safe, convenient, and effective platform for targeted delivery of AB4 to the colorectum. This strategy addresses key limitations of oral dosing and warrants further clinical development for UC and related colorectal inflammatory diseases.

## 1. Introduction

Ulcerative colitis (UC), a chronic immune-mediated inflammatory bowel disease, features a relapsing–remitting course, elevated cancer risk, and poor prognosis [1]. Peak incidence occurs at 20–30 years, imposing significant health and quality-of-life burdens, leading to its WHO designation as a refractory disease [2]. Current therapy relies on pharmacologic agents (5-ASA derivatives, antibiotics, and immunosuppressants) for symptom control [3]. Surgery is also utilized. However, pharmacotherapy often only achieves short-term remission, with relapse remaining a major challenge [4]. Pulsatilla decoction (Baitouweng Tang) is an important formula commonly used clinically for the treatment of UC [5]. Studies have shown that a synergistic effect can be produced by combining Baitouweng Tang with Western medicines, enterolysis, or acupuncture [6,7]. Saponins are the main chemical constituent of Pulsatilla decoction, and anemoside B4 (AB4) is the primary component of Pulsatilla chinensis [8]. Modern pharmacological studies have shown that it has anti-inflammatory, anti-tumor, anti-viral, analgesic, and immunomodulatory effects. In addition, the effectiveness of AB4 in treating UC has been well documented [9,10,11,12], which may be related to its anti-inflammatory, apoptosis-inhibiting, ulcer-protective, and ulcer-repairing effects [13].

Some studies have shown AB4 is a biopharmaceutical classification system (BCS) class III drug that mainly undergoes active transport. The oral administration of these drugs is associated with disadvantages such as poor permeability, low gastrointestinal stability, and poor bioavailability. The rectal route of drug administration may be a viable alternative to oral administration to effectively solve the problems of bioavailability and increase the therapeutic potential of drugs [14]. The rectal mucosa is rich in blood vessels; therefore, a regional or systemic effect is possible while avoiding the first-pass effect, preventing the action of gastric acid and enzymes on the drug, and reducing stimulation [15]. However, there are many limitations in rectal drug delivery, including the low mucosal permeability to macromolecules, limited surface area, and short retention time, which lead to low bioavailability and shorten the time for effective absorption [16]. To overcome these deficiencies, absorption enhancers are usually used to accelerate absorption. Commonly used enhancers include cyclodextrin, surfactants, metal chelators, salicylates, and their derivatives, bioadhesive polymers, etc. These compounds can improve drug absorption either by fluidizing cell membranes to enhance transcellular transport or by reorganizing tight junctions to facilitate paracellular transport [17]. This study aimed to systematically compare the absorption-enhancing effects of several permeation enhancers, such as hydroxypropyl-β-cyclodextrin (HP-β-CD), sodium caprate (SC), chitosan (CS), Tween-80, water-soluble azone (azone), and arginine (Arg), through individual application. The objective was to identify the two most effective enhancers and their optimal concentrations for maximizing AB4 permeation, thereby providing an experimental basis for future development of rectal delivery formulations of AB4.

In the treatment of UC, rectal administration offers advantages, including better cure rates and the rapid relief of symptoms. However, suppositories tend to flow out of the rectum after softening in the body, causing discomfort to patients and reducing compliance [18,19]. Additionally, after rectal administration, nonadhesive suppositories that might reach the end of the colon may undergo the first-pass effect [20,21]. An ideal preparation for rectal administration should be easy to administer without causing any pain during insertion; moreover, it should have a suitable mucoadhesive force so that it does not reach the end of the colon and can avoid the first-pass effect in the liver and gastrointestinal tract [22,23].

In situ gels (ISGs) have solution–gel transition properties, which not only have the advantage of the wide distribution of the enema but also of being quickly gelled at the site of administration to achieve lesion-specific drug delivery locally [24]. The mechanism of formation involves the polymer materials that respond to external stimuli (including temperature, ionic strength, or pH), such that the polymer disperses or polymerization occurs under specific physiological conditions [25,26,27,28]. Poloxamer 407 (P407) has been reported to be well tolerated, with little irritation and sensitization to the skin and mucous membranes, and has been used in intranasal, ocular, and rectal formulations [29,30,31]. With the development of medicinal polymer materials, rectal ISGs show good development and application prospects and have become a research hotspot. This dosage form perfectly combines the advantages of solvents and gels and overcomes the shortcomings of suppositories. It can not only solve the problem of medication compliance in patients but can also avoid the first-pass effect that reduces drug efficacy. Some researchers have achieved good results in the treatment of UC with ISGs [32,33,34].

The aim of this study was to develop AB4-loaded in situ rectal gel incorporated with an optimal absorption enhancer in order to improve AB4’s bioavailability and prolong its retention time. The physicochemical properties of gels were characterized by means of techniques, such as gelation temperature, gel strength, in vitro release, etc. Furthermore, the pharmacokinetic properties and effectiveness of the formulation were evaluated to determine its suitability for rectal delivery, which will provide a reference for the development and clinical application of rectal preparations.

## 2. Materials and Methods

### 2.1. Chemicals, Cell Culture, and Animals

AB4 (purity: 98%) was supplied by Jiangxi BencaoTiangong Technology Co., Ltd. (Nanchang, China). The AB4 reference substance (content: 96.4%) was purchased from the National Institutes for Food and Drug Control (Beijing, China). Poloxamer 407 (P407) and 188 (P188) were purchased from BASF (Ludwigshafen, Germany). Hydroxypropyl methyl cellulose (HPMC) E5 from Jiangxi Alpha High-tech Pharmaceutical Co., Ltd. was a free gift (Nanchang, China). Cell culture reagents, including Dulbecco’s modified Eagle’s medium (DMEM), phosphate-buffered saline (PBS), Hank’s balanced salt solution (HBSS), trypsin–EDTA (0.25%), Cell Counting Kit-8 (CCK-8), interleukin-10 (IL-10), interleukin-1 beta (IL-1β), tumor necrosis factor-alpha (TNF-α), myeloperoxidase enzyme-linked immunosorbent assay (MPO), and lipopolysaccharide activity (LPS), D-lactate (D-LA) ELISA assay kits, were obtained from Beijing Solarbio Science & Technology Co., Ltd. (Beijing, China). HP-β-CD, SC, CS, Tween-80, azone, and Arg were obtained from Xi’an Deli Biochemical Co., Ltd. (Xi’an, China), Shanghai Macklin Biochemical Co., Ltd. (Shanghai, China), Shandong Weikang Biomedical Technology Co., Ltd. (Linyi, China), Sichuan Jinshan Pharmaceutical Co., Ltd. (Meishan, China), Ruicheng Yishan Intermediate Co., Ltd. (Yuncheng, China), and Shanghai Xiehe Amino Acid Co., Ltd. (Shanghai, China), respectively. Fetal bovine serum (FBS) was purchased from Hyclone (Logan, UT, USA), and potassium dihydrogen phosphate and sodium hydroxide were obtained from Xilong Science Co., Ltd. (Shenzhen, China). Methylene blue (MB) was purchased from Shanghai Qingxi Chemical Technology Co., Ltd. (Shanghai, China). Purified water was used after deionization and filtration in a Millipore VR system. Paraformaldehyde (PFA) solution (4%) was purchased from Shanghai Titan Scientific Co., Ltd. (Shanghai, China). Rhamsan gum was provided by Shanghai Yiyang Instrument Co., Ltd. (Shanghai, China). Hematoxylin and eosin stain was purchased from Phygene Biotechnology Co., Ltd. (Fuzhou, China). Sliced paraffin (58–60 °C) was purchased from Sinopharm Chemical Reagent Co., Ltd. (Shanghai, China). Dextran sodium sulfate (DSS) was purchased from Dalian Meilun Biotechnology Co., Ltd. (Dalian, China). Acetonitrile and methanol of high-performance liquid chromatography (HPLC)-grade were purchased from Fisher Scientific (Waltham, MA, USA). All other chemicals were of chromatographic grade and used without further purification.

Sprague-Dawley (SD) rats (with a weight range of 180–200 g) and C57BL/6 mice (with a weight of about 20 g) were supplied by Hunan SJA Laboratory Animal Co., Ltd. (Changsha, China). The animal quality license number was SCXK 2019-0004. All procedures of the animal experiments were in accordance with the Regulations of the Experimental Animal Administration.

Caco-2 cells (National Collection of Authenticated Cell Culture, Shanghai, China) were expanded in 75 cm^2^ flasks (BIOFIL, Guangzhou, China) at 37 °C/5% CO_2_. The cells were maintained in DMEM supplemented with 10% FBS, with medium replenishment every 48 h. At 80–90% confluence, monolayers were harvested via PBS wash and 0.25% trypsin–EDTA digestion (3–5 min; 37 °C). Viable cells were quantified by a hemocytometer using phase-contrast microscopy (Olympus, Tokyo, Japan). For transport studies, passage 30–45 cells were seeded at 2 × 10^5^ cells/cm^2^ on 0.4 μm polycarbonate Transwell^®^ inserts (Corning, New York, NY, USA) in 6-well plates. Monolayer integrity was monitored daily by transepithelial electrical resistance (TEER), with full differentiation achieved within 18–21 days post-seeding. The culture medium was replaced every other day throughout the differentiation period [35].

### 2.2. Screening of Absorption Enhancers

#### 2.2.1. HPLC Method

HPLC (Agilent, Santa Clara, CA, USA) and a Diamonsil 5 µm C 18 column (Dikma, Beijing, China) were used for the determination and quantification of AB4. The mobile phase combined two eluents (A–B) in a 30/70 ratio at a flow rate of 1 mL/min, with detection at 210 nm (eluent A: acetonitrile; eluent B: water; injection volume: 10 μL). The run time was 10 min, and the retention time of AB4 was approximately 7 min. A linear correlation was acquired between the peak area and concentration. The linear equation was y = 1083.5x + 531 (*R*^2^ = 0.9992), where x is the concentration, and y is the peak area. The assay was linear in the concentration range of 10~400 µg/mL.

#### 2.2.2. Screening of Absorption Enhancers Using the Caco-2 Cell

Prior to permeation studies, monolayers were washed three times with prewarmed HBSS (37 ± 1 °C) and equilibrated for 30 min. The apical (AP) chamber received 0.5 mL of HBSS containing AB4 (200 μg/mL) with individual permeation enhancers (2.5% *w*/*v*: HP-β-CD, SC, CS, azone, Tween-80, and Arg). The basolateral (BL) chamber contained 1.5 mL of blank HBSS (n = 6). Plates were incubated at 37 °C/5% CO_2_. The sampling protocol: At predetermined intervals (5, 15, 30, 60, 120, 180, and 240 min), 0.2 mL aliquots were withdrawn from the BL chamber and immediately replaced with fresh HBSS. The samples were centrifuged (10,000 rpm/min; 10 min), with the supernatants analyzed for their AB4 concentration via validated HPLC. The apparent permeability coefficient (Papp) of AB4 was calculated using the following equation:(1)$$\mathrm{Papp}\;=\;dQ/dt\times 1/C_{0}A$$
where dQ/dt is the rate of the AB4 appearance in the receiver side (μg·mL^−1^·min), C_0_ is the initial concentration of the drug in donor solution (μg/mL), and A is the membrane surface area of the Caco-2 monolayer (A = 1.12 cm^2^ in this experiment).

#### 2.2.3. Screening of Absorption Enhancers Using the Franz Diffusion Cell

Fresh porcine rectal specimens (10 cm in length) from a local slaughterhouse were longitudinally incised, inspected for abnormalities, and rinsed with PBS (pH 7.4). Mucosal layers were carefully dissected using forceps, rewashed to remove residual blood, mounted on tin foil, and stored at −20 °C until use [36]. Thawed mucosal membranes were trimmed to >1.13 cm^2^ and mounted between donor and receptor chambers of Franz cells. The receptor compartments contained 37 °C PBS (pH 7.4), with the system temperature maintained at 37 ± 1 °C under 400 rpm magnetic stirring. After 30 min of equilibration, the donor chambers received 1 mL of the test solution containing AB4 (2.1 mg/mL) and 2.5% permeation enhancers (HP-β-CD, SC, CS, azone, Tween-80, and Arg). Aliquots (1 mL) were withdrawn from receptor chambers at 5, 15, 30, 60, 120, 180, and 240 min (n = 6), with immediate PBS replacement to maintain sink conditions. Samples were membrane-filtered (0.22 μm) prior to HPLC quantification of AB4. The cumulative permeation amount (Q_n_) per unit area was calculated using the following equation:(2)$$\sum Q_{n}=(C_{n}V+\sum_{i=1}^{n-1}C_{i}V_{i})/A$$
where Q_n_ is the cumulative permeation of the drug per unit area (μg/cm^2^), C_n_ is the drug concentration at the nth sampling point (μg/mL), V is the volume of the receiving solution in the receiving cell (14.7 mL), V_i_ is the volume of each sampling (mL), C_i_ is the concentration of the receiving fluid at the sampling time, and A is the effective permeation area (1.13 cm^2^).

### 2.3. Preparation of AB4-ISG

AB4-ISG was prepared using the cold method, which leverages the reverse thermal gelation property of P407 to form a low-viscosity solution upon hydration at 4 °C that spontaneously transitions into a semi-solid gel upon temperature increase [37]. AB4-ISGs were fabricated using P407 and P188 as a gel base. The AB4 (2.1% *w*/*v*) and HP-β-CD (2.5%) or SC (1%) were first dissolved in purified water using a magnetic stirrer, followed by the slow addition of various percentages of HPMC (0.44%). With continuous stirring, P407 (17.41%) and P188 (4.07%) were slowly added at room temperature. The prepared gels were kept overnight in the refrigerator to obtain a clear solution.

#### 2.3.1. Gelation Temperature

The gelation temperature was determined using the tilt method [38]. A thermometer was placed in a test tube containing 3 mL of the gel in a thermostatic water bath. The temperature was increased slowly, and the tube was tilted every 30 s to observe whether the contents flowed and to determine gelation. The temperature at which the gel solution did not flow was considered the gelation temperature [39].

#### 2.3.2. Formulation Optimization Based on Box–Behnken Design Response Surface Methodology (BBD-RSM)

BBD-RSM was chosen to optimize the gel formulation as this method required fewer experiments [40]. To screen the major factors for BBD-RSM, a series of single-factor experiments, including the amount of P407, P188, and HPMC, was conducted using the gelling temperature as an index. Based on the results from the single-factor experiments, the formulation of AB4 thermosensitive ISG was optimized using BBD-RSM. In the design phase, the contents of P407 (A), P188 (B), and HPMC (C) were chosen as the independent variables with boundaries: 17.0–20.0% *w*/*w* P407, 4.0–8.0% *w*/*w* P188, and 0.3–0.6% *w*/*w* HPMC. The gelation temperature was used as the dependent variable. The details of the design are shown in Appendix A. A total of 17 experiments were conducted, and the corresponding gelation temperatures were measured.

#### 2.3.3. Model Fitting

The Expert-Design^®^ 8.0.6.1 Software (Minneapolis, MN, USA) was used for statistical analysis. The relationship between independent and dependent variables was analyzed by second-order polynomial regression. The correlation coefficient (*R*^2^) and confidence (*p*) values of the equations were used as the criteria for model fitting [40].

#### 2.3.4. Prediction and Validation of Optimal Formulations

The three-dimensional effect surface map and two-dimensional contour map were drawn based on the quadratic polynomial model, and the influence of the three independent variables, AB4, HP-β-CD, and SC, on the gelation temperature was investigated. With 32 °C chosen as the target gelation temperature, the optimal formulations of the AB4 thermosensitive ISGs were chosen for further analyses. The model prediction ability was evaluated by determining the sol–gel transition temperature of the selected formulations and calculating the deviation. The formulation with the minimum deviation was considered the optimal prescription [41].Deviation = (Predicted value − Observed value)/Predicted value × 100%(3)
where the predicted value was 32 °C, and the observed value was the gelation temperature.

### 2.4. Evaluation of AB4-ISG

#### 2.4.1. Gelation Time

First, 1.5 mL samples of HP-β-CD-AB4-ISG and SC-AB4-ISG were taken in a test tube, equilibrated at 25 °C for 5 min, placed in a water bath at 32 °C, and timed immediately. The change in the sample state was observed, and the time required for the sample to reach the gelation temperature was recorded.

#### 2.4.2. Viscosity, pH, Gelation Strength, and Bioadhesive Force

The pH values of HP-β-CD-AB4-ISG and SC-AB4-ISG were measured using a laboratory pH meter (Mettler Toledo, Budapest, Hungary) at 37 °C in a beaker. The viscosity samples were measured using a digital viscometer (NDJ-9S, Shanghai, China). An appropriate amount of ISGs was placed in a 50 mL beaker. Based on preliminary experiments, a No. 3 rotor and a rotation speed of 12 rpm were chosen for the determination of viscosity at room temperature and the gelation temperature. Adapting Yong et al.’s study [42], 50 g gels in 100 mL graduated cylinders were equilibrated at 37 °C. A 35 g gelation strength tester descent time through a 5 cm gel column was recorded. For intervals of >300 s, minimal incremental weights enabled a 5 cm descent within a 300 s defined gel strength (g). A modified mucoadhesion measurement device was employed to assess the bioadhesive force of the in situ gel [43]. Porcine rectal mucosa was mounted onto two glass vials, which were pre-warmed and maintained at 37 °C to prevent gel hydration. After placing 0.5 mL of the gel between the mucosal layers, one vial was attached to a balance. The bioadhesive force, defined as the detachment stress (dyne/cm^2^), was determined by measuring the minimal weights needed to detach the mucosal interfaces.

#### 2.4.3. Stability

Three samples each of HP-β-CD-AB4-ISG and SC-AB4-ISG were sealed in vials and exposed to various stability testing environments [44,45], including low temperature (4 °C), high temperature (40 °C), high humidity (25 °C; relative humidity: 90 ± 5%), and high-intensity light (4500 lx ± 500 lx). The samples were withdrawn on days 0, 2, 5, and 10 to evaluate changes in their macroscopic appearance, gelation temperature, AB4 content, and whether stratification occurred.

#### 2.4.4. In Vitro Drug Release

AB4’s release kinetics from rectal ISGs (AB4-ISG, HP-β-CD-AB4-ISG, and SC-AB4-ISG) were evaluated using Franz diffusion cells with porcine rectal mucosa (with a 1.13 cm^2^ diffusion area). The receptor chambers contained 14.7 mL of PBS (pH 7.4), while donor chambers received 1 mL of the respective ISG formulation uniformly applied onto the mucosa. The assembly was equilibrated at 37 ± 0.5 °C with 100 rpm agitation for 30 min. Aliquots (0.5 mL) were withdrawn from the receptor compartments at 0.083, 0.25, 0.5, 1, 2, 3, 4, and 6 h, with immediate PBS replacement to maintain sink conditions. Samples were membrane-filtered (0.22 μm) and analyzed by HPLC. The cumulative release rate of AB4 was calculated, and the in vitro release curve was constructed.

### 2.5. Pharmacokinetic Study

#### 2.5.1. UPLC-MS/MS Method

The Shimadzu UPLC system (Shimadzu, Kyoto, Japan) and AB SCIENX 4500 mass spectrometer (Shimadzu, Kyoto, Japan) were used to quantitatively analyze AB4 in rat biological samples. The analyses were conducted on an AB SCIEX QTRAP 4500 triple quadrupole UPLC-MS/MS system (USA) operated in electrospray ionization (ESI) (-) mode. The optimized chromatographic separation of AB4 was performed by using an Ultimate XB-C18 column (50 mm × 2.1 mm; with a 1.8 µm particle size). The mobile phase was a 0.1% formic acid aqueous solution (A) and acetonitrile (B), and the gradient elution procedure was 0–1.2 min, 10% B; 0.1–0.5 min, 10–40% B; 0.5–2.0 min, 40–95% B; 2.0–3.2 min, 95% B; 3.2–4.0 min, 95–10% B; and 4.0–5.0 min, 10% B, with the flow rate maintained at 0.4 mL/min. The injection volume was set at 3 µL, and the column oven was maintained at 35 °C. The most abundant fragment ions in multiple reaction monitoring (MRM) were adopted, and *m*/*z* 1219.5→749.5 for AB4 and *m*/*z* 845.4→637.4 for ginsenoside Rg1 at collision energies of −180 V and −200 V were monitored. The decluster potentials of AB4 and ginsenoside Rg1 were −55 eV and −30 eV, respectively. The total run time of the assay was 5.0 min. The retention time of AB4 was 2.23 min, and the retention time of the ginsenoside Rg1 was 2.22 min. The ion spray voltage was adjusted to 4.0 kV. The common parameters were as follows: the nebulizer gas pressure was 50 psi; the gas temperature was 325 °C.

The linear concentration range of AB4 in rat plasma was 0.01–3.2 µg/mL, with a lower limit of quantification (LLOQ) of 0.01 µg/mL (*R* = 0.9925). The mean AB4 plasma extraction recovery was 90.16 ± 9.83%. The intra-day precision was about 5.80% at the quantitation limit of 300 ng/mL, which provided sufficient sensitivity to characterize pharmacokinetics.

#### 2.5.2. Grouping and Dosing

Thirty Sprague-Dawley rats (maintained for 24 h, given water ad libitum) were randomized into 5 groups: the IV group, wherein 5 mg/kg of AB4 saline solution was administered via the tail vein; the IG group, wherein 200 mg/kg of AB4 aqueous solution was administered via oral gavage; and the rectal groups, where 10.5 mg/kg of AB4-ISG, HP-β-CD-ISG, and SC-ISG was administered anally.

#### 2.5.3. Collection and Treatment of Plasma Samples

Serial blood samples (0.5 mL) were collected at 0.083, 0.25, 0.5, 1, 2, 3, 4, 6, 8, 10, and 24 h post-dosing (the IV group, with an additional 0.03 h timepoint) into heparinized tubes. Plasma was separated by centrifugation (8000 rpm; 10 min; 4 °C) and stored at −80 °C pending bioanalysis. The collected blood samples were placed in preheparinized 1.5 mL anticoagulant centrifuge tubes and centrifuged at 8000× *g* for 10 min at 4 °C. The supernatants were drawn into other centrifuge tubes. Cryopreservation was conducted at −80 °C.

After the plasma samples were thawed at room temperature, 50 µL of plasma, 400 µL of methanol, and 50 µL of an internal standard solution (1 µg/mL of ginsenoside Rg1) were accurately weighed in a centrifuge tube and vortexed for 3 min to precipitate the proteins. After centrifugation at 13,000 rpm for 10 min at 4 °C, the supernatant was filtered through a 0.22 µm membrane, and the plasma drug content was determined according to the above UPLC-MS/MS method. The obtained AB4 plasma concentration results were processed according to the DAS 3.0 program non-compartmental model (Beijing JiDaoChengran Technology Co., Ltd., Beijing, China).

### 2.6. Histopathological Studies

Eighteen SD rats were randomly assigned to the blank control group, HP-β-CD-AB4-ISG group, and SC-AB4-ISG group. Before the experiment, the rectal health of the rats was checked, and no congestion, swelling, or ulcers were observed. The rats were fasted for 12 h prior to the experiment but provided free access to water. During this period, 2 mL of warm, boiled water was drawn and injected into the anuses of the rats, to a distance of 4 cm, to promote the emptying of their feces. This process was performed twice. Rats in the blank group did not receive any treatment. Rats in the HP-β-CD-AB4-ISG and SC-AB4-ISG groups were administered a volume of 0.5 mL of ISGs. After 7 d, each anus was observed for edema, congestion, or discharge of secretions, and rectal tissues were collected and fixed in 4% paraformaldehyde for >24 h, dehydrated, paraffin-embedded, sectioned, stained, and observed using microscopy (Leica DM2500, Wetzlar, Germany).

### 2.7. Rectal Retention Test

Four SD rats were randomly divided into 2 groups. Methylene blue (0.5%) was added to the HP-β-CD-AB4-ISG and SC-AB4-ISG prescriptions. Next, 1 mL of the gel was passed through the anus through a hose attached to the rectum to a distance of 4 cm above the anus, as indicated in a previously published protocol [33]. The hair color around the anus was observed at 0.5, 3, and 6 h after administration to check for leakage. The rats were sacrificed, and the distribution and adhesion of HP-β-CD-AB4-ISG and SC-AB4-ISG in the rectum were analyzed.

### 2.8. Pharmacodynamic Studies

A total of 72 male C57BL/6J mice were randomized into nine cohorts (n = 8/group): the control, DSS model, positive control, and high/low-dose AB4-ISG, HP-β-CD-ISG, and SC-AB4-ISG groups. Following 1 week of acclimatization and 24 h of fasting, colitis was induced via 7 days of 3% DSS in drinking water in all non-control groups. Concurrently initiated treatments comprised daily rectal administration (100 μL; 3–4 cm insertion depth) of saline (control/model groups) or respective ISG formulations (treatment groups), followed by 15 s inverted. The daily stool and rectal bleeding in mice were scored as the degree of the disease activity index (DAI) according to the human fecal occult blood test–benzidine method as follows: normal stool—0, soft stool—1, unformed stool—2, and loose stool—3; blood-negative—0, blood-positive—1, stool with blood—2, and blood in stool—3; and no change in weight rate—0, 1–5%—1, 5–10%—2, 10–15%—3, and exceeds 15%—4. The three scores were added up to assess the DAI.

On day 8, mice were sacrificed for sample collection. The spleens were excised, weighed, and subjected to statistical analysis. Entire colorectal segments were harvested for length measurement and morphological assessment. Blood samples were collected, clotted at room temperature for 2 h, and centrifuged (13,000 rpm, 15 min, and 4 °C) to obtain serum, which was stored at −80 °C until quantification of LPS and D-Lac levels using assay kits. Colon segments located 10 cm proximal to the anus were homogenized in PBS. After centrifugation, the resultant supernatant was analyzed by ELISA to determine the levels of IL-10, IL-1β, and TNF-α, as well as MPO activity, according to the respective kit protocols. Additionally, colon tissues were fixed in 4% paraformaldehyde for 24 h, processed through standard histological procedures (dehydration, paraffin embedding, sectioning, and staining), and examined by bright-field microscopy for morphological evaluation.

## 3. Results and Discussion

### 3.1. Results of Absorption Enhancers Screening

In this study, Caco-2 cells and Franz diffusion cells were employed to screen for suitable absorption enhancers and their optimal concentrations, with the aim of promoting the absorption of B4 and improving its bioavailability.

#### 3.1.1. Caco-2 Cell Permeation Studies

As shown in Figure 1a, all five absorption enhancers promoted AB4 absorption, among which HP-β-CD had the best effect, whereas SC, azone, Arg, and Tween-80 had comparable absorption effects. As the safety of SC is already known, we continued to investigate the absorption effects of HP-β-CD and SC at different concentrations. Combined with the CCK-8 method, it was found that 2.5% HP-β-CD and 1% SC had the best absorption effect (Figure 1b,c) and that the cell survival rate was almost 100%, indicating low toxicity (Figure 1d,e).

#### 3.1.2. Franz Diffusion Cell

In order to verify the effects of the Caco-2 cell experiments, this experiment continued to investigate the effects of the six absorption enhancers (HP-β-CD, SC, CS, azone, Tween-80, and Arg) using the Franz diffusion cell. All six absorption enhancers exhibited different degrees of absorption effects (Figure 2a), among which HP-β-CD had the best pro-absorption effect, whereas SC and Tween-80 had comparable effects. SC is widely used as a food additive and can also be used as an absorption enhancer in suppositories, owing to its good safety profile. Accordingly, the corresponding HP-β-CD and SC were selected as absorption enhancers, and their absorption effects at different concentrations were analyzed. As shown in Figure 2b,c, 2.5% HP-β-CD and 1% SC, respectively, had the best absorption effects.

AB4 is a large-molecular-weight compound (1221.38 Da) classified as a BCS III drug, which is likely absorbed via the paracellular pathway or through enhanced membrane fluidity. This study employed Caco-2 cell permeability assays and Franz diffusion cell experiments to screen suitable absorption enhancers from among HP-β-CD, SC, CS, Tween-80, azone, and Arg. Among them, HP-β-CD not only improves drug solubility but also interacts with cholesterol and phospholipids in the cell membrane, transiently and reversibly altering membrane structure and fluidity to enhance transcellular permeability [46]. SC reversibly opens tight junctions, thereby creating paracellular pathways to promote the absorption of hydrophilic drugs. This process is reversible and typically recovers within hours. Furthermore, SC has a long history of human use with food additive status and has been extensively evaluated as an intestinal permeation enhancer for the oral delivery of macromolecules [47]. CS also reversibly opens epithelial tight junctions, though its mechanism is more associated with direct interaction with junctional proteins to facilitate paracellular transport [46]. Tween-80, a non-ionic surfactant, inserts into the lipid bilayer of cell membranes, disrupting lipid organization and increasing membrane fluidity, thereby reducing resistance to transcellular drug diffusion [48]. The positively charged Arg can bind to the negatively charged head groups of membrane phospholipids through electrostatic interactions, perturbing membrane structure [49]. The results from both Caco-2 cell permeability studies and Franz diffusion cell experiments demonstrated that HP-β-CD and SC were the most effective absorption enhancers, promoting AB4 absorption by either increasing membrane fluidity or opening tight junctions between cells, which aligns with their known mechanisms of action. Based on comprehensive consideration, 2.5% HP-β-CD and 1% SC were selected as absorption enhancers for AB4-ISG.

### 3.2. Optimization of the ISG Formulation

Single-factor experiments revealed that the amounts of P407 and P188 had a greater influence on the gelation temperature than the amount of HPMC. The gelation temperature decreased with an increase in the P407 concentration (Figure 3a). However, with an increase in the P188 concentration, the gelation temperature increased initially and then decreased (Figure 3b), whereas with an increase in the HPMC concentration, the gelation temperature only decreased slightly, indicating little change (Figure 3c). Consequently, the amounts of P188, P407, and HPMC were chosen as variables for the optimization process. The variable A represents the proportion of P407 (17–20%), B represents the proportion of P188 (4–8%), C represents the proportion of HPMC (0.3–0.6%), and T/Tg represents the calculated gelation temperature. The variables A, B, and C were designed at three levels: −1, 0, and 1. The results for the gelation temperature are shown in Appendix A. The quadratic polynomial regression used to indicate the relationship between the independent and dependent variables was carried out using Design-Expert 10 software (version 8.0.6.1). The goodness of fit (*R*^2^) and confidence (*p*) of the fitting equation were used as criteria for model evaluation. The quadratic polynomial equation was as follows: T = 30.50 − 5.20 × A + 2.36 × B + 0.00875 × C − 0.14 × AB − 0.35 × AC + 0.5 × BC (*p* < 0.0001; *R*^2^ = 0.9757). The *R*^2^ and *p*-values indicate that the model was a good fit.

Three-dimensional effect surface diagrams were created using the Design-Expert 8.0.6.1 software based on the above quadratic polynomial. As shown in Figure 4a–f, the gelation temperature decreased with an increase in the P407 concentration but increased with an increase in the P188 concentration. HPMC had no obvious effect on the gelation temperature, whereas P407 exhibited the greatest influence on the gelation temperature. The effect of AB4 and the absorption enhancers HP-β-CD and SC on the gelation temperature was approximately 4 °C. Therefore, three alternative formulations with a gelation temperature of 32 °C were selected to verify the established quadratic polynomial model to ensure that the gel was liquid at room temperature and gelled rapidly at rectal temperature, which normally ranges from 36.32 to 37.76 °C in humans. As shown in Table 1, using 32 °C as the target temperature, three prescriptions were screened, the deviation in the predicted value from the measured value was calculated, and the optimal prescription was determined based on the smallest deviation to be 17.41% P407, 4.07% P188, and 0.44% HPMC, and an appropriate amount of water.

### 3.3. AB4-ISG Characterization

#### 3.3.1. Gelation Temperature, Gelation Time, pH, and Viscosity

HP-β-CD-AB4-ISG and SC-AB4-ISG appeared as clear and transparent liquids. The gelation temperature should be consistent with the rectal temperature. The shorter the gelation time, the slower the drug loss, and the less likely it is that the burst release reaction occurs. The gelation temperatures of HP-β-CD-AB4-ISG and SC-AB4-ISG were 32.83 ± 0.06 °C and 35.93 ± 0.15 °C, consistent with the rectal temperature. The gelation times were 28.00 ± 4.00 s and 63.33 ± 5.51 s. The pH values were 6.77 and 8.5, and there was no need to add a pH regulator, as it met the requirements of pH 7~8 for rectal administration. The viscosity at 25 °C was approximately 400 mPa·s, and the ISGs’ viscosity at 37 °C was approximately 4500 mPa·s. Preparations with a higher viscosity increase the residence time of the drug in the rectum, thereby enhancing drug absorption and reducing the possibility of burst release.

#### 3.3.2. Gelation Strength and Bioadhesive Force

The two most important indices of gel degree are strength and bioadhesive force. The presence of both suitable gel strength and bioadhesion is a key determinant for effective drug delivery, as they promote prolonged localization at the rectal site and improve the rectal absorption efficiency. Bioadhesion, defined as the binding force between the gel and the rectal mucosa at physiological temperature, is a critical determinant for rectal retention. Sufficient adhesion retains the gel at the administration site, preventing its displacement to the colon and bypassing the first-pass effect. The results showed that HP-β-CD-AB4-ISG and SC-AB4-ISG exhibited a suitable gel strength (50 ± 5 g) and bioadhesion (11.82 ± 0.45 dyne/cm^2^). The gels were easily administered, retained at the site without leakage, and their adhesion properties were sufficient to prevent colon migration, bypassing first-pass metabolism [50].

#### 3.3.3. Stability Analysis

The stability test results are shown in Table 2. The results showed that the content of SC-AB4-ISG decreased at a high temperature. It was stable under low-temperature, high-humidity, and high-intensity light conditions. Therefore, these formulations should be stored at low temperatures, indicating that both HP-β-CD-AB4-ISG and SC-AB4-ISG were suitable for low-temperature storage.

#### 3.3.4. In Vitro Drug Release Analysis

The in vitro drug release is shown in Figure 5, where the release curves of the three formulations coincide, demonstrating a similar slow-release behavior and achieving continuous drug release. Each ISG preparation released approximately 80% of the drug in 6 h. Three different release models were used to fit the release profiles of ISG, including zero-order, first-order, and Higuchi kinetic models. The model equations are shown in Appendix A, where “t” represents the corresponding time, “M” signifies the percentage of cumulative release, and *R*^2^ denotes the coefficient of determination, which is used to determine the most suitable model [51]. The cumulative release rate data of HP-β-CD-AB4-ISG and SC-AB4-ISG conformed to the first-order model. The in vitro dissolution linear correlation coefficient *R^2^* values were 0.9836 and 0.9784, indicating that the Fickian diffusion fit the in vitro drug release of HP-β-CD-AB4-ISG and SC-AB4-ISG better, and the fitting result was good. The drug was released from the gel by diffusion.

### 3.4. Rectal Retention Test and Histopathological Studies

After the rectal administration of HP-β-CD-AB4-ISG and SC-AB4-ISG containing methylene blue dye, its retention in the rectum was observed. After 0.5 h of administration, it became a dark blue gel. After 6 h, the blue gradually faded. There was no obvious leakage in each period after administration, and they were distributed in the intestinal segment about 2~13 cm above the anus, with a wide distribution area. These findings indicated that the ISG was retained in the rectum for at least 6 h, thereby ensuring its release (Appendix A).

Safety tests were performed to check for irritation or damage to rectal tissues in rats after the rectal administration of HP-β-CD-AB4-ISG and SC-AB4-ISG. No abnormalities to the rectal tissue were noted by the naked eye. The stained histopathological sections revealed that the local mucosal epithelium of the rectum in the blank control group, HP-β-CD-AB4-ISG group, and SC-AB4-ISG group was intact. No obvious inflammatory cell infiltration, edema, or ulcers were noted (Figure 6), indicating that HP-β-CD-AB4-ISG and SC-AB4-ISG did not irritate or damage the rectum.

### 3.5. Pharmacokinetic Study Analysis

The mean plasma concentration versus time curve is shown in Figure 7, and the main pharmacokinetic parameters are shown in Appendix A. After the administration of a 200 mg/kg AB4 solution in rats via the IG route, the peak time (T_max_) was 0.542 ± 0.246 h, indicating more rapid absorption after IG administration. Compared with that of the IV injection group, the peak concentration (C_max_) was lower, and the absolute bioavailability was 0.16% for the IG group. Compared with that in the IG group, the T_max_ and half-life (t_1/2_) in the AB4-ISG group were prolonged, and the absolute bioavailability was 38.45%, which was significantly higher than 0.16%. These findings indicated that the formulation of AB4 into a rectal ISG could significantly increase the drug-retention time in the rectum, thereby increasing its absorption and bioavailability. The C_max_ for the HP-β-CD-AB4-ISG and SC-AB4-ISG groups increased significantly, and the absolute bioavailabilities were 72.05% and 60.34%, respectively, which increased by 1.88 and 1.57 times compared with those for the AB4-ISG group. These findings showed that the addition of appropriate concentrations of absorption enhancers to ISG formulations can increase the extent of drug absorption. Therefore, rectal ISGs can improve the absorption of AB4. Moreover, ISG formulations containing appropriate absorption enhancers can increase drug absorption by nearly two-fold compared with ISGs without absorption enhancers.

### 3.6. Pharmacodynamic Studies Analysis

#### 3.6.1. Body Weight and DAI Scores

Body weight and the DAI are vital for evaluating drug efficacy in anti-UC. Body weight reflects the drug’s impact on the overall health of mice, while the DAI score serves as an indicator of UC severity [52]. The body weights of mice in the blank group increased slowly without any abnormal changes (Figure 8). The body weights of mice in the model group decreased versus those of mice in the normal control group (*p* < 0.01). Moreover, the body weights of mice increased after AB4-ISG, HP-β-CD-AB4-ISG, and SC-AB4-ISG intervention compared with those of mice in the model group. The weight loss of mice with HP-β-CD-AB4-ISG and SC-AB4-ISG did not change much compared with that of mice in the positive control and AB4-ISG groups. These results showed that AB4-ISG, HP-β-CD-AB4-ISG, and SC-AB4-ISG could attenuate the tendency of body weight loss in mice with UC, and that the effect was the best with high doses of HP-β-CD-AB4-ISG and SC-AB4-ISG. As shown in Table 3, the DAI score of mice in the blank group was 0, and no significant change in body weight or any bloody stool was noted. The DAI score of the model group significantly increased compared with that of the blank control group (*p* < 0.01). The DAI score of mice with UC decreased after AB4-ISG, HP-β-CD-AB4-ISG, and SC-AB4-ISG intervention compared with that of model mice (*p* < 0.05), suggesting that administration of the ISGs could alleviate the symptoms of UC. The DAI score of the HP-β-CD-AB4-ISG and SC-AB4-ISG group decreased compared with that of the positive drug group and the AB4-ISG group. These findings showed that ISG administration could improve the DAI score.

#### 3.6.2. Spleen Weight and Colon Length

Spleen weight and colon length in the UC model are indicators of intestinal inflammatory damage. As shown in Figure 9a, the average spleen weight increased significantly in the model group compared with that in the blank group (*p* < 0.001), suggesting a systemic inflammatory response and immune dysfunction. The spleen weight was reduced in the AB4-ISG, HP-β-CD-AB4-ISG, SC-AB4-ISG, and positive groups compared with that in the model group (*p* < 0.01). Moreover, the spleen weight was significantly lower in the HP-β-CD-AB4-ISG and SC-AB4-ISG groups compared with that in the AB4-ISG group (*p* < 0.001). As shown in Figure 9b, the whole bowel was significantly atrophied and inflamed in the model group (*p* < 0.001). However, mice in the HP-β-CD-AB4-ISG and SC-AB4-ISG groups exhibited better recovery of the whole bowel than those in the AB4-ISG or model group (*p* < 0.05), where the bowel length was comparable to that of healthy mice. These findings suggested that AB4-ISGs may ameliorate splenomegaly and colonic damage by alleviating inflammatory responses and modulating immune function [53].

#### 3.6.3. Serum Indicators

The intestinal mucosa is a key factor in the development of UC. When the intestinal mucosal barrier is compromised, bacteria and their metabolites invade the intestines and bloodstream, resulting in an increase in D-Lac and LPS levels. As shown in Figure 10a,b, LPS and D-Lac levels in the model group were significantly elevated by 95.30% and 46.67%, respectively (*p* < 0.01), compared with those in the blank group, suggesting impairment of the intestinal mucosal barrier in mice with UC. LPS and D-Lac activity in the AB4-ISG, HP-β-CD-AB4-ISG, and SC-AB4-ISG groups decreased (*p* < 0.05) compared with that in the model group. These results indicated that AB4-ISGs ameliorate DSS-induced intestinal mucosal damage in UC mice.

#### 3.6.4. Colon Tissue Inflammatory Factor and MPO Activity

Proinflammatory/anti-inflammatory cytokine imbalance critically contributes to UC pathogenesis (Figure 11). Secretion of the pro-inflammatory factors TNF-α and IL-1β in the model group was highly significantly elevated (*p* < 0.01), and the secretion of the anti-inflammatory factor IL-10 was highly significantly reduced (*p* < 0.01) compared with that in the blank group. The levels of the pro-inflammatory factors TNF-α and IL-1β in the AB4-ISG, HP-β-CD-AB4-ISG, and SC-AB4-ISG groups were highly significantly elevated (*p* < 0.05), and the level of the anti-inflammatory factor IL-10 was highly significantly reduced (*p* < 0.05) compared with those in the model group. These results indicate that AB4-ISGs effectively inhibit the release of inflammatory mediators and alleviate intestinal inflammation.

MPO activity in the inflamed colon is an important indicator that reflects neutrophil infiltration (Figure 11d). The colons of mice from the model group showed higher MPO levels (*p* < 0.001) compared with those of mice from the blank group. In contrast, MPO levels in the AB4-ISG, HP-β-CD-AB4-ISG, and SC-AB4-ISG groups were significantly decreased compared with the positive group (*p* < 0.05). These findings suggested that AB4-ISGs can ameliorate the intestinal oxidative stress state in UC mice.

#### 3.6.5. Histopathological Examination of Colonic Tissue

Histopathological examination staining (Figure 12) revealed significant treatment-dependent preservation of colonic cytoarchitecture in UC mice. The blank group exhibited structurally intact mucosa with well-preserved crypt organization, uniform glandular distribution, and compact submucosa devoid of pathology. The model group tissues demonstrated extensive crypt destruction, epithelial denudation, severe submucosal/lamina propria edema, and dense inflammatory infiltrates. The AB4-ISG treatment partially ameliorated mucosal damage, reduced edema, and substantially mitigated inflammatory infiltrates. Critically, the HP-β-CD-AB4-ISG and SC-AB4-ISG formulations demonstrated superior mucosal preservation relative to AB4-ISG monotherapy, with significantly reduced epithelial necrosis, indicating enhanced restitution of colonic tissue.

## 4. Conclusions

The rectal route is a useful mode for the delivery of pharmacological agents, enabling targeted drug delivery to inflamed mucosa while circumventing hepatic first-pass effects [54]. While conventional enemas and suppositories suffer from poor patient compliance, ISGs are a potentially useful vehicle for the incorporation and delivery of biological therapeutics for delivery to inflamed mucosal surfaces. To overcome the inherent absorption limitations of AB4, we pioneered the incorporation of permeation enhancers into ISG formulations. This study established, for the first time, that HP-β-CD and SC significantly potentiate AB4 absorption. The optimized HP-β-CD-AB4-ISG and SC-AB4-ISG formulations demonstrated critical pharmaceutical attributes: physiologically relevant sol–gel transitions occurred at 32.83 ± 0.06 °C and 35.93 ± 0.15 °C, respectively, while in vitro release kinetics revealed sustained drug delivery (~80% cumulative release within 6 h). Significantly enhanced bioavailability was achieved through prolonged rectal retention.

We engineered novel thermosensitive ISGs incorporating HP-β-CD or SC as permeation enhancers for rectal AB4 delivery. Comprehensive characterization confirmed optimal gelation properties, controlled release kinetics, and significantly enhanced bioavailability. These findings validate HP-β-CD-AB4-ISG and SC-AB4-ISG as promising rectal delivery platforms, particularly for drugs undergoing extensive first-pass metabolism, hepatotoxic compounds, and biologics with poor membrane permeability. Future studies will explore clinical translation potential and the mechanistic depth of these formulations.

## Figures and Tables

**Figure 1 pharmaceutics-17-01400-f001:**
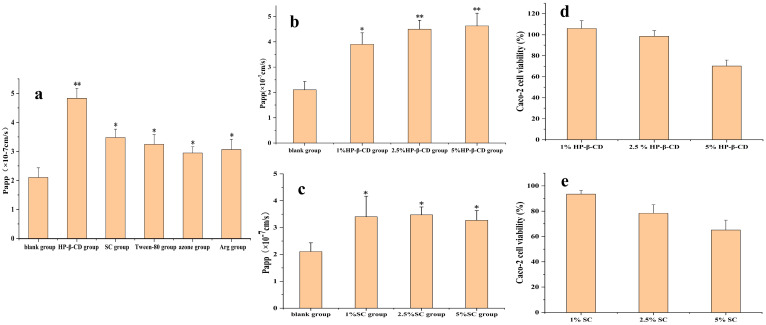
(**a**) Screening of absorption enhancer types in AB4 transport in the Caco-2 cell monolayer. (**b**) Screening of HP-β-CD concentrations in AB4 transport in the Caco-2 cell monolayer. (**c**) Screening of SC concentrations in AB4 transport in the Caco-2 cell monolayer. (**d**) Effect of HP-β-CD concentrations on cytotoxicity. (**e**) Effect of SC concentrations on cytotoxicity. Compared with the blank group, * *p* < 0.1 and ** *p* < 0.05.

**Figure 2 pharmaceutics-17-01400-f002:**
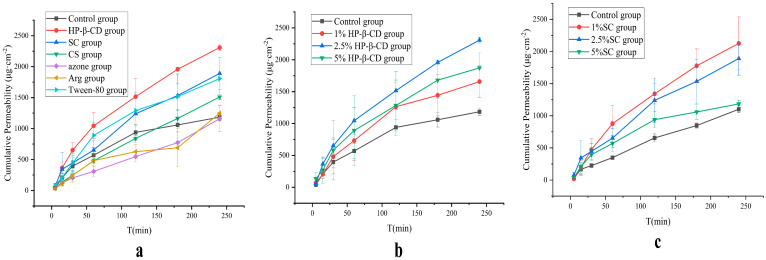
(**a**) Screening of absorption enhancer types in AB4 permeability through porcine rectal mucosa. (**b**) Screening of HP-β-CD concentrations in AB4 permeability through porcine rectal mucosa. (**c**) Screening of SC concentrations in AB4 permeability through porcine rectal mucosa.

**Figure 3 pharmaceutics-17-01400-f003:**
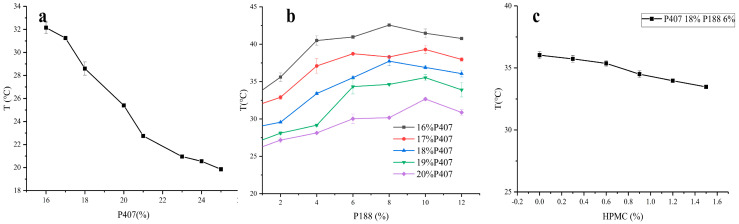
Effects of P407, P188, and HPMC contents on gelation temperature (**a**–**c**).

**Figure 4 pharmaceutics-17-01400-f004:**
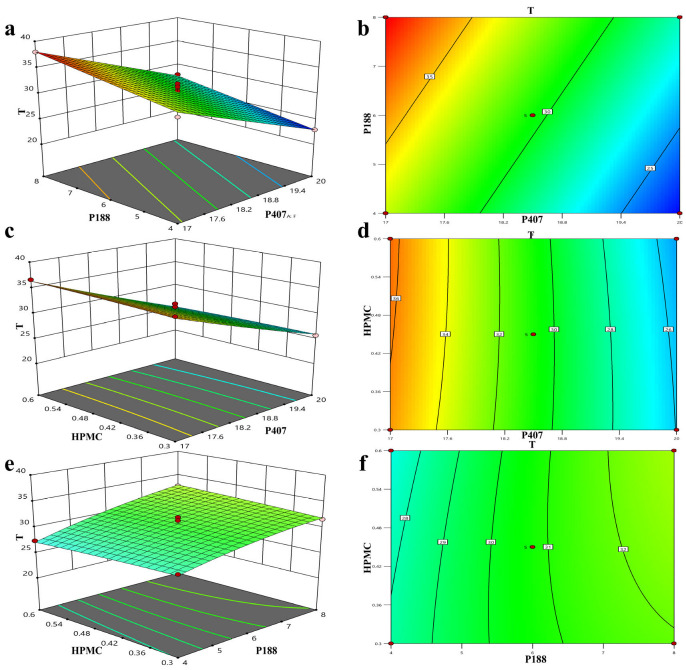
Three-dimensional effect surface diagram of AB4 in situ gels plotted using Design-Expert 8.6.0.1 software according to quadratic polynomial model (**a**–**f**).

**Figure 5 pharmaceutics-17-01400-f005:**
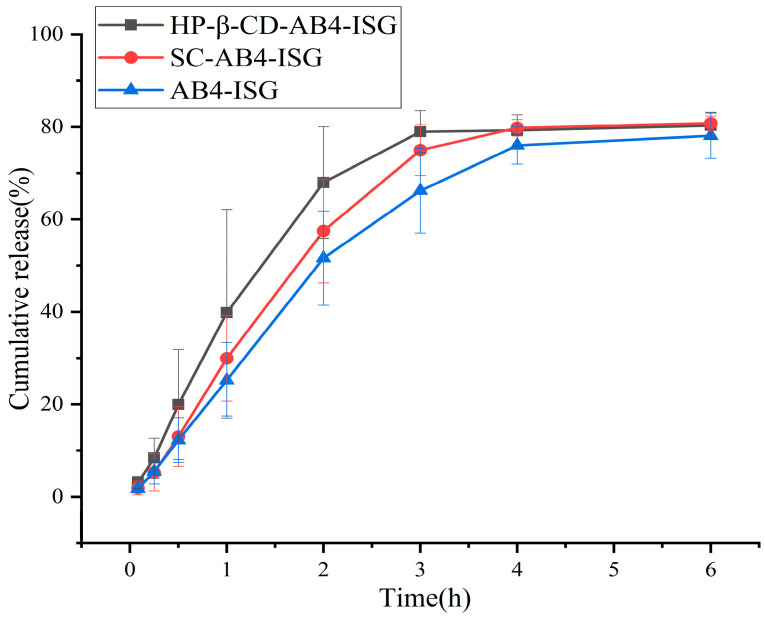
In vitro drug release profiles of AB4 from different formulations in phosphate buffer (n = 6).

**Figure 6 pharmaceutics-17-01400-f006:**
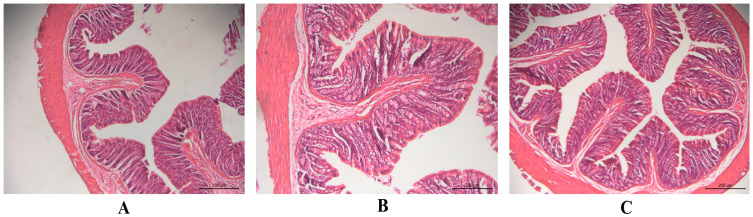
The pathological changes in the rectal tissue of rats in each group after rectal administration for 7 d (×200 nm): (**A**) blank control group, (**B**) HP-β-CD-AB4-ISG group, and (**C**) SC-AB4-ISG group rectal administration.

**Figure 7 pharmaceutics-17-01400-f007:**
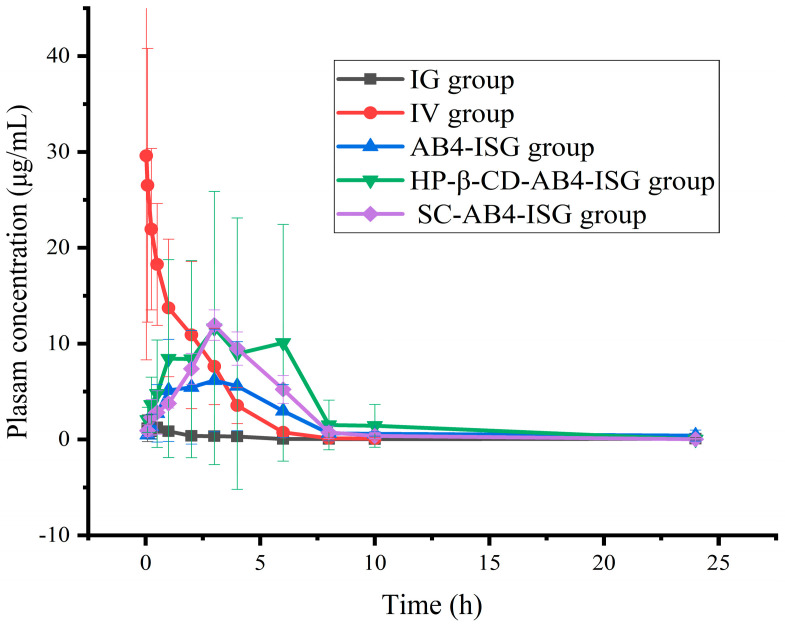
Concentrations of components in plasma samples of each group after administration of AB4 preparation (n = 6).

**Figure 8 pharmaceutics-17-01400-f008:**
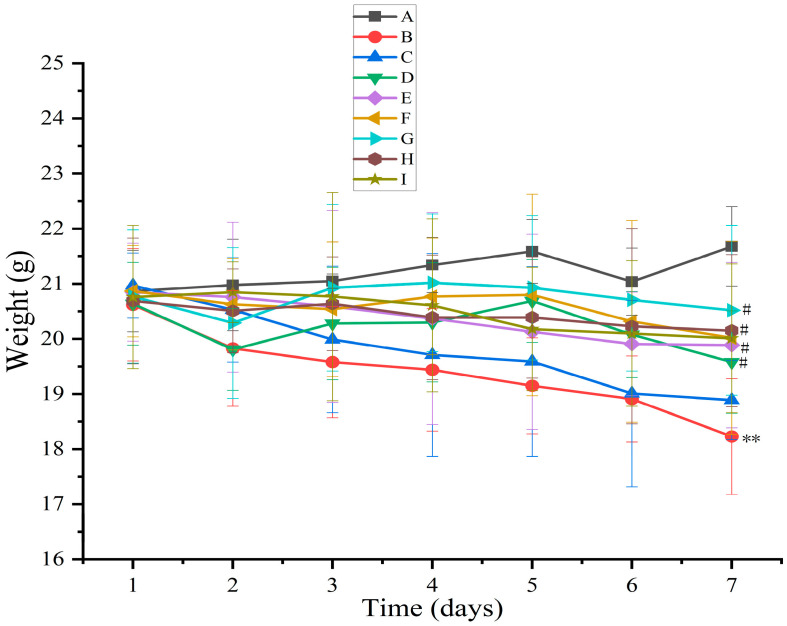
Effect on the body weight of mice. **A**: Blank group, **B**: model group, **C**: positive group, **D**: AB4-ISG low-dose group, **E**: AB4-ISG high-dose group, **F**: HP-β-CD-AB4-ISG low-dose group, **G**: HP-β-CD-AB4-ISG high-dose group, **H**: SC-AB4-ISG low-dose group, and **I**: SC-AB4-ISG high-dose group. Compared with the blank group, ** *p* < 0.01; compared with the positive group, # *p* < 0.05.

**Figure 9 pharmaceutics-17-01400-f009:**
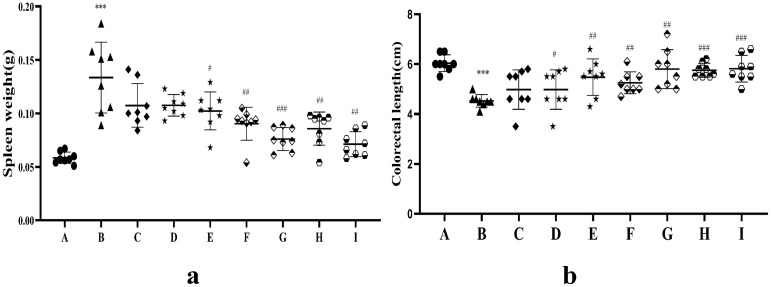
(**a**) Spleen weight and (**b**) colon length after treatment.●**A**: Blank group, ▲**B**: model group, ◆**C**: positive group, ★**D**: AB4-ISG low-dose group, ★**E**: AB4-ISG high-dose group, 
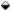
**F**: HP-β-CD-AB4-ISG low-dose group, 
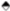
**G**: HP-β-CD-AB4-ISG high-dose group, 

**H:** SC-AB4-ISG low-dose group, and 
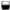
**I**: SC-AB4-ISG high-dose group. Compared with the blank group, *** *p* < 0.001; compared with the positive group, # *p* < 0.05, ## *p* < 0.01, and ### *p* < 0.001.

**Figure 10 pharmaceutics-17-01400-f010:**
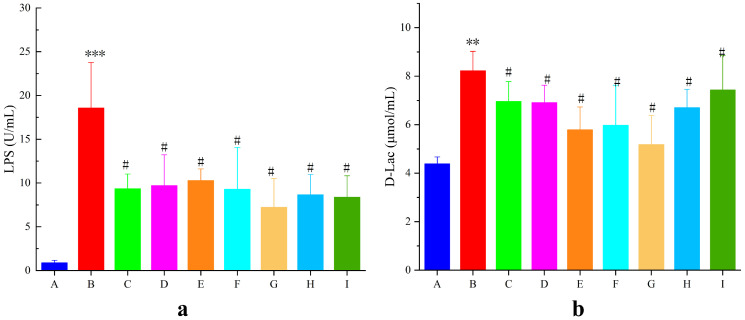
Effects on (**a**) LPS and (**b**) D-Lac levels in mice with UC levels. **A**: Blank group, **B**: model group, **C**: positive group, **D**: AB4-ISG low-dose group, **E**: AB4-ISG high-dose group, **F**: HP-β-CD-AB4-ISG low-dose group, **G**: HP-β-CD-AB4-ISG high-dose group, **H:** SC-AB4-ISG low-dose group, and **I**: SC-AB4-ISG high-dose group. Compared with the blank group, ** *p* < 0.01, and *** *p* < 0.001; compared with the positive group, # *p* < 0.05.

**Figure 11 pharmaceutics-17-01400-f011:**
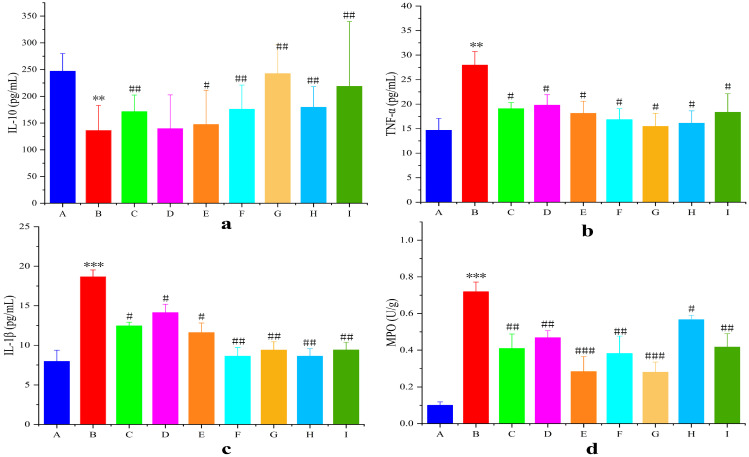
Statistical mean fluorescence intensity of (**a**) IL-10, (**b**) TNF-α, (**c**) IL-1β, and (**d**) MPO. **A**: Blank group, **B**: model group, **C**: positive group, **D**: AB4-ISG low-dose group, **E**: AB4-ISG high-dose group, **F**: HP-β-CD-AB4-ISG low-dose group, **G**: HP-β-CD-AB4-ISG high-dose group, **H:** SC-AB4-ISG low-dose group, and **I**: SC-AB4-ISG high-dose group. Compared with the blank group, ** *p* < 0.01, and *** *p* < 0.001; compared with the positive group, # *p* < 0.05, ## *p* < 0.01, and ### *p* < 0.001.

**Figure 12 pharmaceutics-17-01400-f012:**
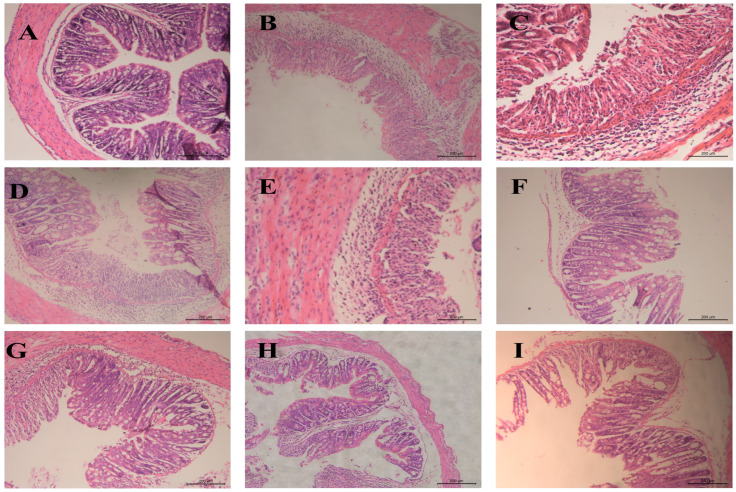
The typical histopathological examination staining photos of the colon section. (**A**): Blank group, (**B**): model group, (**C**): positive group, (**D**): AB4-ISG low-dose group, (**E**): AB4-ISG high-dose group, (**F**): HP-β-CD-AB4-ISG low-dose group, (**G**): HP-β-CD-AB4-ISG high-dose group, (**H**): SC-AB4-ISG low-dose group, and (**I**): SC-AB4-ISG high-dose group.

**Table 1 pharmaceutics-17-01400-t001:** Validation of the prescriptions.

Formulation	P407%	P188%	HPMC%	Prediction of Gel Temperature (°C)	Measured Gel Temperature (°C)	Deviation (%)
1	17.41	4.07	0.44	32	32.10 ± 0.10	0.31
2	17.67	4.84	0.34	32	33.43 ± 0.10	4.38
3	17.50	4.34	0.31	32	33.36 ± 0.26	4.06

**Table 2 pharmaceutics-17-01400-t002:** The results of stability.

	Sampling Time (d)			0	2	5	10
State							
High temperature	Appearance			Clear and transparent gel			
	Gelation temperature/°C	HP-β-CD		/	/	/	/
		SC		/	/	/	/
	Content/%	HP-β-CD		100%	96.94 ± 1.54	96.03 ± 0.96	95.35 ± 0.33
		SC		100%	93.81 ± 0.32	85.72 ± 1.05	79.57 ± 0.87
	Stratification			No			
Low temperature	Appearance			Clear and transparent liquid			
	Gelation temperature/°C	HP-β-CD		32.77 ± 0.15	32.79 ± 0.03	31.96 ± 0.53	32.03 ± 0.96
		SC		35.87 ± 0.15	35.74 ± 0.52	35.32 ± 0.57	35.63 ± 1.12
	Content/%	HP-β-CD		100%	99.94 ± 0.12	99.03 ± 0.85	98.87 ± 0.39
		SC		100%	99.81 ± 0.51	99.12 ± 0.94	98.93 ± 0.89
	Stratification			No			
High humidity	Appearance			Clear and transparent liquid			
	Gelation temperature/°C	HP-β-CD		33.05 ± 0.43	32.45 ± 0.54	32.07 ± 0.96	32.85 ± 0.43
		SC		36.25 ± 0.87	35.56 ± 0.85	35.07 ± 1.57	35.96 ± 0.35
	Content/%	HP-β-CD		100%	99.28 ± 0.54	98.32 ± 0.69	98.53 ± 1.57
		SC		100%	99.32 ± 0.38	98.93 ± 0.89	98.01 ± 0.23
	Stratification			No			
High-intensity light	appearance			Clear and transparent liquid			
	Gelation temperature/°C		HP-β-CD	32.65 ± 0.05	32.58 ± 0.13	33.23 ± 0.86	32.98 ± 0.52
			SC	35.82 ± 0.35	35.56 ± 0.85	35.02 ± 0.32	34.73 ± 1.29
	Content/%		HP-β-CD	100%	99.75 ± 1.03	99.54 ± 0.23	99.21 ± 1.57
			SC	100%	98.51 ± 1.58	98.28 ± 0.59	97.34 ± 1.04
	Stratification			No			

**Table 3 pharmaceutics-17-01400-t003:** Disease activity index scores. **A**: Blank group, **B**: model group, **C**: positive group, **D**: AB4-ISG low-dose group, **E**: AB4-ISG high-dose group, **F**: HP-β-CD-AB4-ISG low-dose group, **G**: HP-β-CD-AB4-ISG high-dose group, **H:** SC-AB4-ISG low-dose group, and **I**: SC-AB4-ISG high-dose group. Compared with the blank group, * *p* < 0.05, ** *p* < 0.01, and *** *p* < 0.001; compared with the positive group, # *p* < 0.05, ## *p* < 0.05.

Days	A	B	C	D	E	F	G	H	I
1	0	0	0	0	0	0	0	0	0
2	0	1.00 ± 1.00	0.67 ± 0.58	0.67 ± 0.58	0.67 ± 0.58	0.67 ± 0.58	0.67 ± 0.58	0.67 ± 0.58	0.67 ± 0.58
3	0	1.33 ± 1.15 *	1.33 ± 0.58	1.00 ± 1.00	1.33 ± 0.58	1.33 ± 1.15	1.00 ± 0.00	0.67 ± 0.58 **#	0.67 ± 0.58 **#
4	0	2.67 ±1.00 **	1.67 ± 0.58 **	1.67 ± 0.58 **#	1.67 ± 0.58 **#	1.67 ± 0.58 **#	1.67 ± 0.58 **#	1.33 ± 0.58 ***##	1.33 ± 0.58 ***##
5	0	3.10 ± 0.34 **	2.67 ± 0.58 *	2.33 ± 1.15 *#	2.67 ± 0.58 *#	2.33 ± 1.15 **#	2.67 ± 0.58 **#	1.33 ± 0.58 ***##	1.53 ± 0.67 ***##
6	0	3.52 ± 0.93 **	2.80 ± 1.73 **	2.42 ± 1.53 **#	2.70 ± 1.73 **#	2.67 ± 1.53 **#	2.33 ± 1.15 **#	2.33 ± 1.15 **#	1.67 ± 1.09 ***#
7	0	3.67 ± 0.58 **	3.03 ± 1.15 *	2.67 ± 1.15 **#	2.83 ± 1.15 **#	2.37 ± 1.15 **#	2.53 ± 1.73 **#	2.33 ± 1.15 **#	2.33 ± 1.15 **#

## Data Availability

The original contributions presented in this study are included in the article/Appendix A. Further inquiries can be directed to the corresponding authors.

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
