# Peer review of "Anemoside B4 Rectal Thermosensitive In Situ Gel to Treat Ulcerative Colitis by Overcoming Oral Bioavailability Barriers with Absorption Enhancer-Assisted Delivery"

_pharmaceutics, 2025, doi:10.3390/pharmaceutics17111400_

Round 1

Reviewer 1 Report

Comments and Suggestions for Authors

This research article investigates the development and evaluation of anemoside B4 rectal thermosensitive in situ gels for the treatment of ulcerative colitis. The authors aimed to overcome the poor oral bioavailability and epithelial permeability of AB4 by formulating it into a rectal ISG with the aid of absorption enhancers. The core novelty of this work lies in the successful formulation of AB4 into a thermosensitive in situ gel for rectal administration, specifically incorporating absorption enhancers (HP-β-CD and SC) to improve its bioavailability.

Comments to authors

  1. The details about the used HPLC method of analysis are not mentioned anywhere in the manuscript. A brief description would be sufficient
  2. In section 2.3., “AB4-ISG was prepared using the cold method”, please add a reference.
  3. In section 2.3.2, the underlying paragraph is in bold. Please ensure consistent formatting.
  4. In section 2.7, the reference 33 was added in different format than others in the text.
  5. Although HP-β-CD and SC were identified as effective absorption enhancers, the precise mechanisms by which they improve AB4 permeation were not fully elucidated. While it’s known that HP-β-CD can form inclusion complexes and SC can transiently open tight junctions, a more detailed investigation would provide deeper insights.
  6. On what basis the boundaries: P407 17.0%-20.0% w/w, P188 4.0%-8.0% w/w, and HPMC 0.3%-0.6% w/w were chosen. Just a simple clarification is needed.
  7. Why the target gelation temperature was set at 32OC and how this temperature would affect the ease of administration of the dosage form if the room temperature is high enough up to this level.
  8. The study focuses on the immediate properties and efficacy of the ISGs. Information regarding the long-term physical and chemical stability of the formulations (e.g., shelf-life, storage conditions, potential degradation products of AB4 within the gel) is crucial for practical application.
  9. The study compares the ISG formulations to oral AB4 and a positive control (unspecified in the methods, but likely a standard UC treatment). A direct comparison with other established rectal UC treatments (e.g., mesalazine enemas or suppositories) would provide a clearer picture of the relative advantages and disadvantages of the developed ISGs.
  10. While short-term rectal tolerability was assessed, long-term biocompatibility and potential for chronic irritation or adverse effects with repeated administration were not fully explored. This is particularly relevant for a chronic condition like UC. Elaborate more on the histological analysis after prolonged administration, and potentially include markers of inflammation or tissue damage to ensure long-term safety.
  11. The study mentions mucoadhesion as a desirable property for rectal formulations but does not provide quantitative data on the mucoadhesive strength or interaction with the mucus layer in vivo. While retention was observed, direct evidence of mucoadhesion would be valuable. Implement in vitro or ex vivo methods to quantify the mucoadhesive properties of the ISGs, such as tensile strength measurements or rotating cylinder methods, to support the observed prolonged retention.
  12. While the goal is local delivery, some systemic absorption is inevitable. A more detailed analysis of systemic exposure and potential off-target effects of AB4 delivered via ISG would be beneficial, especially if higher doses are considered.
  13. Discuss the potential differences in rectal physiology between rodents and humans and how these might impact the translation of the findings. Consider ex vivo human rectal tissue studies if feasible.
  14. Ensure all statistical analyses are clearly reported, including the specific tests used, p-values, and confidence intervals for all comparisons. For example, in Table 2, the asterisks for significance are relative to the blank group, but comparisons to the model group are also crucial and should be clearly indicated or presented.
  15. Some figures, particularly Figure 1 & Figure 5, contain many sub-panels. While informative, ensuring all labels are clearly legible and the data points are easily distinguishable would enhance presentation. Consider splitting into multiple figures if necessary, so that labels are more readable without the need to zoom in.
  16. The caption of Figure 3 needs to include the definition of the three formulas in the description.

Addressing these points would significantly enhance the scientific depth, translational potential, and overall impact of the research.

Comments on the Quality of English Language

The manuscript is generally well-written and conveys the scientific information clearly. However, there are several instances of grammatical errors, awkward phrasing, and minor typos that require attention to improve the overall readability and professional presentation of the article. Below are some examples and general observations:

  1. Some sentences are overly long or contain convoluted structures, which could be simplified for better clarity. For instance, in the abstract, “through overcome oral bioavailability barriers with enhancer-assisted delivery” could be rephrased for better flow.
  2. There are occasional missing commas, incorrect use of hyphens/dashes, and inconsistent spacing around punctuation marks.
  3. A few instances of imprecise word choice or slightly informal language were noted. For example, “overcome oral bioavailability barriers” could be improved to “overcoming oral bioavailability barriers.”
  4. Minor typos and spelling mistakes are present throughout the text.
  5. Overall, the English is understandable, but a careful review and revision are necessary to meet the high standards of a scientific publication like Pharmaceutics.

Author Response

29, September 2025

Reviewer

Pharmaceutics

Dear Reviewer,

Thank you very much for handling our manuscript entitled “Anemoside B4 rectal thermosensitive in situ gel to treat ulceraive colitis through by overcominge oral bioavailability barriers with absorption enhancer -assisted delivery” (pharmaceutics- 3877623) and providing us with valuable comments and suggestions. We have carefully read the comments and suggestions from the reviewers and editor and then revised our manuscript accordingly. We would also like to respond to these comments and suggestions one by one below:

Comment 1:

The details about the used HPLC method of analysis are not mentioned anywhere in the manuscript. A brief description would be sufficient.

Our reply:

Thank you for this suggestion. According to your suggestion, we have added the content of the HPLC method of analysis in the manuscript.

HPLC (Agilent, USA) and a Diamonsil 5 µm C 18 column (Dikma, China) was used for the determination and quantification of AB4. The mobile phase combined two eluents (A:B) in a 30/70 ratio at a flow rate of 1 mL/min, with detection at 210 nm. Eluent A: acetonitrile, eluent B: water. Injection volume: 10 μL. The run time was 10min and the retention time of AB4 was approximately 7 min. A linear correlation was acquired between peak area and concentration. The linear equation was y=1083.5x+531 (R2=0.9992), where x is the concentration and y is the peak area. The assay was linear in the concentration of 10~400 µg/mL.

Comment 2:

In section 2.3., “AB4-ISG was prepared using the cold method”, please add a reference.

Our reply:

Thank you for this question. According to your suggestion, we have added a reference in the manuscript.

Comment 3:

In section 2.3.2, the underlying paragraph is in bold. Please ensure consistent formatting.

Our reply:

Thank you for pointing out this. We have ensured consistent formatting in the manuscript.

Comment 4:

In section 2.7, the reference 33 was added in different format than others in the text.

Our reply:

Thank you for pointing out this. We have revised the reference 33 format in the text.

Comment 5:

Although HP-β-CD and SC were identified as effective absorption enhancers, the precise mechanisms by which they improve AB4 permeation were not fully elucidated. While it’s known that HP-β-CD can form inclusion complexes and SC can transiently open tight junctions, a more detailed investigation would provide deeper insights.

Our reply:

Thank you for pointing out this. In this study, both the Caco-2 cell permeation assay and the Franz diffusion cell model demonstrated that HP-β-CD and SC significantly enhanced the permeation of AB4. According to literature, HP-β-CD not only improved drug solubility but also bound to the mucosa, inducing disorganization of the phospholipid bilayer and facilitating pore formation, thereby promoting mucosal drug absorption [1]. SC, which has a long history of human use and holds food additive status, has been widely evaluated as an intestinal permeation enhancer for the oral delivery of macromolecules. Furthermore, the cellular damage caused by SC to enterocytes is reversible, recovering rapidly after discontinuation of use, indicating low toxicity [2]. AB4 is a large molecular weight compound (1,221.38 Da, BCSclass III drug) that is likely absorbed via the paracellular pathway. We evaluated the mechanism using TEER measurements across Caco-2 cell monolayers. The results demonstrated that both HP-β-CD and SC reversibly opened tight junctions between cells. This observation supports their role in facilitating paracellular transport, consistent with the mechanism described above.

[1] Ghadiri M, Young PM, Traini D. Strategies to Enhance Drug Absorption via Nasal and Pulmonary Routes. Pharmaceutics. 2019,11, 113.

[2] Tran H, Aihara E, Mohammed FA, Qu H, Riley A, Su Y, Lai X, Huang S, Aburub A, Chen JJH, Vitale OH, Lao Y, Estwick S, Qi Z, ElSayed MEH. In Vivo Mechanism of Action of Sodium Caprate for Improving the Intestinal Absorption of a GLP1/GIP Coag-onist Peptide. Mol Pharm. 2023, 20, 929-941.

Comment 6:

On what basis the boundaries: P407 17.0%-20.0% w/w, P188 4.0%-8.0% w/w, and HPMC 0.3%-0.6% w/w were chosen. Just a simple clarification is needed.

Our reply:

Thank you for pointing out this. Based on the preliminary experimental results, an increase in P407 concentration led to a decrease in the gelation temperature: when the concentration was below 15%, the gelation temperature exceeded 50℃, while at 20%, gelation occurred rapidly at room temperature. Thus, the initial concentration range for P407 was selected as 16%-20%. Subsequent experiments involving the addition of P188 (2%-10%) showed that the gelation temperature first increased and then decreased with rising P188 concentration. Accordingly, the optimized ranges for P407 and P188 were narrowed to 17%-20% and 4%-8%, respectively. Further experiments with a fixed P407/P188 ratio (18%/6%) and varying HPMC concentrations (0.3%-1.5%) revealed that increasing HPMC slightly reduced the gelation temperature, but exceeding 0.6% resulted in non-uniform dissolution or even phase separation. Therefore, the final concentration range for HPMC was determined to be 0.3%-0.6%.

Comment 7:

Why the target gelation temperature was set at 32℃ and how this temperature would affect the ease of administration of the dosage form if the room temperature is high enough up to this level.

Our reply:

Thank you for pointing out this. Based on previous studies, the absorption promoter (SC) was found to increased the gelation temperature by approximately 4°C (Table 1). Considering that the rectal temperature is around 37°C, the target gelation temperature was set at 32°C to ensure rapid gelation after administration. If the ambient temperature exceeds rectal temperature, pre-gelation may occur in the syringe. To avoid injection difficulties, it is recommended to cool the formulation in a refrigerator (2-8°C) for 5-10 min prior to administration to restore its low-viscosity state.

Table 1  The gelation temperature of in situ gel formulation AB4 and absorption enhancers

P407/%

P188/%

HPMC/%

AB4/%

HP-β-CD/%

SC/%

gelation temperature/℃

1

17

4.7

0.5

/

/

/

35.07±0.15

2

17

4.7

0.5

0.42

/

/

33.00±0.56

3

17

4.7

0.5

0.42

2.5

/

36.50±0.53

4

17

4.7

0.5

0.42

/

1

39.10±0.56

Comment 8:

The study focuses on the immediate properties and efficacy of the ISGs. Information regarding the long-term physical and chemical stability of the formulations (e.g., shelf-life, storage conditions, potential degradation products of AB4 within the gel) is crucial for practical application.

Our reply:

Thank you for this valuable comment. Data on the stability of both HP-β-CD- AB4-ISG and SC-AB4-ISG have been incorporated in the manuscript. The results showed that the content of SC-AB4-ISG decreased at a high temperature. It is stable under low temperature, high humidity and high-intensity light conditions. Therefore, these formulations should be stored at low temperatures, indicating that both HP-β-CD-AB4-ISG and SC-AB4-ISG were suitable for low-temperature storage.

Table 2. The results of stability.

Sampling time(d)、State、Condition

0

2

5

10

High

temperature

appearance

clear and transparent gel

gelation

temperature/℃

HP-β-CD

/

/

/

/

SC

/

/

/

/

content/%

HP-β-CD

100%

96.94±1.54

96.03±0.96

95.35±0.33

SC

100%

93.81±0.32

85.72±1.05

79.57±0.87

whether stratification

No

Low

temperature

appearance

clear and transparent liquid

gelation

temperature/℃

HP-β-CD

32.77±0.15

32.79±0.03

31.96±0.53

32.03±0.96

SC

35.87±0.15

35.74±0.52

35.32±0.57

35.63±1.12

content/%

HP-β-CD

100%

99.94±0.12

99.03±0.85

98.87±0.39

SC

100%

99.81±0.51

99.12±0.94

98.93±0.89

whether stratification

No

High

humidity

appearance

clear and transparent liquid

gelation

temperature/℃

HP-β-CD

33.05±0.43

32.45±0.54

32.07±0.96

32.85±0.43

SC

36.25±0.87

35.56±0.85

35.07±1.57

35.96±0.35

content/%

HP-β-CD

100%

99.28±0.54

98.32±0.69

98.53±1.57

SC

100%

99.32±0.38

98.93±0.89

98.01±0.23

whether stratification

No

High

intensity light

appearance

clear and transparent liquid

gelation

temperature/℃

HP-β-CD

32.65±0.05

32.58±0.13

33.23±0.86

32.98±0.52

SC

35.82±0.35

35.56±0.85

35.02±0.32

34.73±1.29

content/%

HP-β-CD

100%

99.75±1.03

99.54±0.23

99.21±1.57

SC

100%

98.51±1.58

98.28±0.59

97.34±1.04

whether stratification

No

Comment 9:

The study compares the ISG formulations to oral AB4 and a positive control (unspecified in the methods, but likely a standard UC treatment). A direct comparison with other established rectal UC treatments (e.g., mesalazine enemas or suppositories) would provide a clearer picture of the relative advantages and disadvantages of the developed ISGs.

Our reply:

Thank you for pointing out this. In the present study, the oral AB4 was selected as a control to validate the advantage of AB4-ISG in overcoming the limitations of oral administration and enhancing bioavailability. Regarding the positive control, instead of directly employing a commercially available mesalazine suppository, mesalazine was formulated into ISG at a prescribed dosage. This approach was designed to minimize formulation-related variables and, concurrently, to resolve the challenge of dose adaptation in the animal model.

Comment 10:

While short-term rectal tolerability was assessed, long-term biocompatibility and potential for chronic irritation or adverse effects with repeated administration were not fully explored. This is particularly relevant for a chronic condition like UC. Elaborate more on the histological analysis after prolonged administration, and potentially include markers of inflammation or tissue damage to ensure long-term safety.

Our reply:

Thank you for this valuable comment. We fully agree that long-term safety is a decisive factor for the clinical translation of any formulation intended for chronic conditions like UC. The 7d rectal safety assessment conducted in our study was designed as an initial evaluation of local mucosal tolerability following short-term exposure. The results demonstrated no significant mucosal irritation or tissue damage, indicating a favorable preliminary safety profile. We completely acknowledge that short-term studies cannot fully predict the risks associated with prolonged administration. As rightly emphasized by the reviewer, systematic long-term safety evaluation is essential. Therefore, we have formally planned to conduct a comprehensive GLP compliant repeated dose toxicity study as a key next step.

Comment 11:

The study mentions mucoadhesion as a desirable property for rectal formulations but does not provide quantitative data on the mucoadhesive strength or interaction with the mucus layer in vivo. While retention was observed, direct evidence of mucoadhesion would be valuable. Implement in vitro or ex vivo methods to quantify the mucoadhesive properties of the ISGs, such as tensile strength measurements or rotating cylinder methods, to support the observed prolonged retention.

Our reply:

Thank you for pointing out this. We have added the data measuring mucoadhesive properties to the manuscript.

A modified mucoadhesion measurement device was employed to assess the bioad-hesive force of the in situ gel [3]. Porcine rectal mucosa was mounted onto two glass vials, which were pre-warmed and maintained at 37 °C to prevent gel hydration. After placing 0.5 mL of the gel between the mucosal layers, one vial was attached to a balance. The bioadhesive force, defined as the detachment stress (dyne/cm²), was determined by measuring the minimal weights needed to detach the mucosal interfaces.

The two most important indexes of gel degree are strength and bioadhesive force. The presence of both suitable gel strength and bioadhesion are key determinants for effective drug delivery, as they promote prolonged localization at the rectal site, and improve the rectal absorption efficiency. Bioadhesion, defined as the binding force between the gel and the rectal mucosa at physiological temperature, is a critical determinant for rectal retention. Sufficient adhesion retains the gel at the administration site, preventing its displacement to the colon and bypassing the first-pass effect. The results showed that HP-β-CD-AB4-ISG and SC-AB4-ISG exhibit suitable gel strength (50±5 g) and bioadhesion (11.82±0.45 dyne/cm²). The gels were easily administered, retained at the site without leakage, and their adhesion properties are sufficient to prevent colon migration, bypassing first-pass metabolism

[3] Seo YG, Kim DW, Yeo WH, Ramasamy T, Oh YK, Park YJ, Kim JA, Oh DH, Ku SK, Kim JK, Yong CS, Kim JO, Choi HG. Docetaxel-loaded thermosensitive and bioadhesive nanomicelles as a rectal drug delivery system for enhanced chemothera- peutic effect. Pharm Res. 2013, 30, 1860-1870.

Comment 12:

While the goal is local delivery, some systemic absorption is inevitable. A more detailed analysis of systemic exposure and potential off-target effects of AB4 delivered via ISG would be beneficial, especially if higher doses are considered. Discuss the potential differences in rectal physiology between rodents and humans and how these might impact the translation of the findings. Consider ex vivo human rectal tissue studies if feasible.

Our reply:

Thank you for this valuable comment. The rectal administration route offers dual therapeutic value, enabling both local and systemic drug delivery. In this study, systemic absorption was achieved through the rectal inferior vena cava pathway, effectively bypassing hepatic first-pass metabolism to attain therapeutic efficacy against UC. In current rectal drug delivery research, rodent models have become the most widely used in vivo system for evaluating drug release/absorption characteristics and pharmacodynamics. Their advantages include manageable breeding, controlled genetic backgrounds, and sufficient physiological similarity in gastrointestinal basal structure to humans [4], serving to minimize risks in subsequent clinical trials. We appreciate the reviewer's constructive suggestion. We have planned to conduct permeation and retention studies using ex vivo human rectal tissue in our subsequent work, pending ethical approval and clinical resource coordination, to obtain more compelling data.

[4] DeSesso JM, Jacobson CF. Anatomical and physiological parameters affecting gastrointestinal absorption in humans and rats. Food Chem Toxicol. 2001, 39, 209-228.

Comment 13:

Ensure all statistical analyses are clearly reported, including the specific tests used, p-values, and confidence intervals for all comparisons. For example, in Table 2, the asterisks for significance are relative to the blank group, but comparisons to the model group are also crucial and should be clearly indicated or presented.

Our reply:

Thank you for this question. We have revised Table 2 in the manuscript.

Comment 14:

Some figures, particularly Figure 1 & Figure 5, contain many sub-panels. While informative, ensuring all labels are clearly legible and the data points are easily distinguishable would enhance presentation. Consider splitting into multiple figures if necessary, so that labels are more readable without the need to zoom in.

Our reply:

Thank you for this question. We have revised in the manuscript.

Comment 15:

The caption of Figure 3 needs to include the definition of the three formulas in the description.

Our reply:

Thank you for this question. We have added he definition of the three formulas in the manuscript.

Comment 16:

Comments on the Quality of English Language

The manuscript is generally well-written and conveys the scientific information clearly. However, there are several instances of grammatical errors, awkward phrasing, and minor typos that require attention to improve the overall readability andprofessional presentation of the article. Below are some examples and general observations:

Some sentences are overly long or contain convoluted structures, which could be simplified for better clarity. For instance, in the abstract, “through overcome oral bioavailability barriers with enhancer-assisted delivery” could be rephrased for better flow.

There are occasional missing commas, incorrect use of hyphens/dashes, and inconsistent spacing around punctuation marks.

A few instances of imprecise word choice or slightly informal language were noted. For example, “overcome oral bioavailability barriers” could be improved to “overcoming oral bioavailability barriers.”

Minor typos and spelling mistakes are present throughout the text.

Overall, the English is understandable, but a careful review and revision are necessary to meet the high standards of a scientific publication like Pharmaceutics.

Our reply:

Thank you for this question. We have revised in the manuscript.

I hope the above responses and the revised manuscript can satisfy the questions and

comments from the reviewers and you. Should you have any queries, please do not

hesitate to contact me at (+86) 0791-87119623. Thanks for your consideration of our

work. I look forward hearing from you.

Yours Sincerely,

Guo-Song Zhang

Corresponding Author:

Prof. Guo-Song Zhang, National Pharmaceutical Engineering Center of Solid Preparation in Chinese Herbal Medicine, Jiangxi University of Chinese Medicine,  Nanchang 330006, China.

Tel: (+86) 0791-87119623

E-mail: zhgs81411@aliyun.com

Reviewer 2 Report

Comments and Suggestions for Authors

Anemoside B4 rectal thermosensitive in situ gel to treat ulcerative colitis through overcome oral bioavailability barriers with enhancer- assisted delivery

The study consists of developing thermosensitive rectal in‑situ gels (ISGs) of Anemoside B4 (AB4), the main saponin from Pulsatilla chinensis, to overcome its poor oral permeability and low bioavailability for ulcerative colitis (UC). Using 2.5 % hydroxypropyl‑β‑cyclodextrin or 1.0 % sodium caprate as absorption enhancers together with 17.41 % poloxamer 407, 4.07 % poloxamer 188 and 0.44 % HPMC, the researchers obtained gels that met all physicochemical requirements, caused no rectal irritation, remained in the rectum for an extended period, and substantially increased AB4 bioavailability, resulting in marked improvement of UC in an animal model. Thus, the enhancer‑assisted, poloxamer‑based rectal ISG platform offers a safe, convenient and effective means of delivering AB4 to the colorectum, addressing the key limitations of oral dosing.

Although the topic is interesting for this journal, several issues have been identified during the review process, as outlined below:

Comment 1- The main limitation of the manuscript is that the authors have not thoroughly discussed or validated most of the results and characterization data by analyzing their implications and comparing them with findings from other studies. I recommend that the authors present the results in a clear and concise manner and then discuss their significance and advantages in relation to the intended applications.

Additionally, every figure in the manuscript is overloaded with information, uses an excessively small font, and appears blurry, which hampers interpretation of the results. I strongly recommend reorganizing the figures, splitting them where appropriate, and enlarging the text to enhance clarity and provide a cleaner visual presentation of the data.

Comment 2- Introduction section:

  • It is recommended to define abbreviations before using them; for example, P407, P188 and HPMC.
  • Discuss the appropriate absorption enhancers, explaining which ones were chosen and the rationale behind their selection.
  • I suggest the authors rewrite the last paragraph of the Introduction section to clearly state the objectives, mention the hypothesis, and highlight the significance of the study.

Comment 3- Materials and methods:

  • For the chemical components, include the chemical name in addition to the commercial name.
  • It is recommended to define abbreviations before using them; for example, DMEM, FBS, 3SS, DSS, LPS and D-Lac.
  • Preparation of AB4-ISG: Please provide a more detailed description of how the various formulations were prepared, including the exact quantities of reagents and solvents used, and explain what the “cold method” entails.
  • Model Fitting: As a reviewer, I recommend that the authors expand the statistical methods section by specifying the exact version of Model Fitting Expert‑Design®, the configuration parameters (e.g., convergence criteria, maximum iterations), and whether any data transformations or variable‑selection procedures were applied before fitting a second‑order (quadratic) polynomial regression. Please also report additional fit diagnostics beyond R² and p‑values (e.g., residual analysis, standard error of estimate, cross‑validation results) and provide sample size and descriptive statistics for each variable. Finally, correct typographical errors: replace “twice polynomial regression” with “second‑order polynomial regression,” use the proper notation R², and refer to “p‑values” rather than “confidence (P) values.”
  • In vitro drug release: Describe the HPLC method, specifying the column type, instrumentation, and operating conditions, as well as the calibration curve, detection wavelength, and the concentration range evaluated during validation.
  • Pharmacokinetic study: How were the blood samples analyzed? Please describe the analytical method employed.
  • Rectal retention test: Please be consistent and format the bibliography uniformly throughout the document (reference 33).

Comment 4- Results and Discussion:

  • Please elaborate further on the sub-sections of the Results and Discussion section, providing appropriate references to support the claims and findings. Proper citation of relevant studies is essential to strengthen the validity of the results. I strongly recommend that the authors discuss and validate most of the results and characterization data by analyzing their implications and comparing them with findings from other studies.
  • Optimization of the ISG formulation: Please explain the rationale behind this statement: “HPMC had no obvious effect on gelation temperature, whereas P407 exhibited the greatest influence on gelation temperature. The effect of AB4 and the absorption enhancers HP-β-CD and SC on the gelation temperature was approx-imately 4°C. “
  • Gelation temperature, gelation time pH, viscosity and gelation strength: Do all formulations exhibit identical viscosity and gelation strength, and what are the implications of these results?
  • In vitro drug release: What are the implications of all formulations fitting a first‑order release model?
  • It is recommended to define abbreviations before using them; for example, H&E.
  • Please improve the clarity of the figures by increasing the font size of all labels, legends, and axis titles.

Comment 5- Please italicize the terms insituinvivo and invitro everywhere they appear in the manuscript, including in the title, subtitles, and reference list.

Comment 6- Please improve the overall presentation of the manuscript: number the lines, use a uniform font size throughout the text, insert proper spacing between numbers and their units, avoid excessive bold formatting, and attend to other typographic details.

Comments on the Quality of English Language

Finally, the English is sub‑par and the manuscript lacks clarity and logical flow, which makes it difficult for readers to follow the argument and remain engaged.

Author Response

29, September 2025

Reviewer

Pharmaceutics

Dear Reviewer,

Thank you very much for handling our manuscript entitled “Anemoside B4 rectal thermosensitive in situ gel to treat ulceraive colitis through by overcominge oral bioavailability barriers with absorption enhancer-assisted delivery” (pharmaceutics- 3877623) and providing us with valuable comments and suggestions. We have carefully read the comments and suggestions from the reviewers and editor and then revised our manuscript accordingly. We would also like to respond to these comments and suggestions one by one below:

Comment 1:

The main limitation of the manuscript is that the authors have not thoroughly discussed or validated most of the results and characterization data by analyzing their implications and comparing them with findings from other studies. I recommend that the authors present the results in a clear and concise manner and then discuss their significance and advantages in relation to the intended applications.

Our reply:

Thank you for pointing out this. We have expanded the discussion of research findings in the manuscript

Comment 2:

Additionally, every figure in the manuscript is overloaded with information, uses an excessively small font, and appears blurry, which hampers interpretation of the results. I strongly recommend reorganizing the figures, splitting them where appropriate, and enlarging the text to enhance clarity and provide a cleaner visual presentation of the data.

Our reply:

Thank you for this question. We have improved the resolution of the figures and adjusted the font sizes in the manuscript to enhance overall clarity.

Comment 3:

It is recommended to define abbreviations before using them; for example, P407, P188 and HPMC.

Our reply:

Thank you for this question. We have revised in the manuscript.

Comment 4:

Discuss the appropriate absorption enhancers, explaining which ones were chosen and the rationale behind their selection.

Our reply:

Thank you for pointing out this. We have added this section to the manuscript.

Some studies have shown AB4 is a biopharmaceuticals classification system (BCS) class III drug that mainly undergoes active transport. The oral administration of these drugs is associated with disadvantages such as poor permeability, low gastrointestinal stability, and poor bioavailability. The rectal route of drug administration may be a vi-able alternative to oral administration to effectively solve the problems of bioavailability and increase the therapeutic potential of drugs [14]. The rectal mucosa is rich in blood vessels; therefore, a regional or systemic effect is possible while avoiding the first-pass effect, preventing the action of gastric acid and enzymes on the drug, and reducing stimulation [15]. However, there are many limitations in rectal drug delivery, including the low mucosal permeability to macromolecule, limited surface area and short retention time, which leads to low bioavailability and shortens the time for effective absorption [16]. To overcome these deficiencies, absorption enhancers are usually used to accelerate absorption. Commonly used enhancers include cyclodextrin, surfactants, metal chelators, salicylates and their derivatives, and bioadhesive polymers, which can either improve transcellular transport by fluidizing the cell membranes or paracellular transport by rearranging the tight junctions [17]. However, there are few researches on the application of other absorption enhancers in rectal drug delivery.

Comment 5:

I suggest the authors rewrite the last paragraph of the Introduction section to clearly state the objectives, mention the hypothesis, and highlight the significance of the study.

Our reply:

Thank you for pointing out this. We have rewritten the last paragraph of the Introduction section in the manuscript.

Comment 6:

For the chemical components, include the chemical name in addition to the commercial name. It is recommended to define abbreviations before using them; for example, DMEM, FBS, 3SS, DSS, LPS and D-Lac.

Our reply:

Thank you for this question. We have revised in the manuscript.

Comment 7:

Preparation of AB4-ISG: Please provide a more detailed description of how the various formulations were prepared, including the exact quantities of reagents and solvents used, and explain what the “cold method” entails.

Our reply:

Thank you for this question. We have revised in the manuscript.

AB4-ISG was prepared using the cold method, which leverages the reverse thermal gelation property of P407 to form a low-viscosity solution upon hydration at 4 °C that spontaneously transitions into a semi-solid gel upon temperature increase [37]. AB4-ISGs were fabricated using P407and P188 as a gel base. The AB4 (2.1 % w/v) and HP-β-CD (2.5 %) or SC (1 %) were first dissolved in purified water using a magnetic stirrer followed by the slow addition of various percentages of HPMC (0.44 %). With continuous stirring, P407 (17.41 %) and P188 (4.07 %) were slowly added at room temperature. The prepared gels were kept overnight in the refrigerator to obtain a clear solution.

Comment 8:

Model Fitting: As a reviewer, I recommend that the authors expand the statistical methods section by specifying the exact version of Model Fitting Expert‑Design®, the configuration parameters (e.g., convergence criteria, maximum iterations), and whether any data transformations or variable‑selection procedures were applied before fitting a second‑order (quadratic) polynomial regression. Please also report additional fit diagnostics beyond R² and p‑values (e.g., residual analysis, standard error of estimate, cross‑validation results) and provide sample size and descriptive statistics for each variable. Finally, correct typographical errors: replace “twice polynomial regression” with “second‑order polynomial regression,” use the proper notation R², and refer to “p‑values” rather than “confidence (P) values.”

Our reply:

Thank you for this question. The software used was Expert-Design® (Version 8.0.6.1, Stat-Ease Inc., Minneapolis, MN). Model fitting was performed using default configuration parameters: convergence tolerance set to 1×10⁻⁸, maximum iterations set to 100, significance level α = 0.05, and variance inflation factor (VIF) ≤ 10. Prior to fitting the second-order polynomial regression, no data transformations or variable selection procedures were applied to retain all complete model terms. Additional fit diagnostics have been supplemented, with results shown in Table 1. Other requested revisions have been incorporated into the manuscript.

Table 1. Regression coefficient and variance analysis of each factor

Error source

SS

f

S

F

p

Significance

Model

262.18

6

43.70

66.90

<0.0001

**

A-P407

215.90

1

215.90

330.53

<0.0001

**

B-P188

44.70

1

44.70

68.43

<0.0001

**

C-HPMC

6.125×10-4

1

6.125×10-4

9.377×10-4

0.9762

AB

0.078

1

0.078

0.12

0.7362

AC

0.49

1

0.49

0.75

0.4067

BC

1.01

1

1.01

1.55

0.2421

ABC

0.000

0

Residual

6.53

10

0.65

Lack of Fit

2.77

6

0.46

0.49

0.7920

Pure Error

3.76

4

0.94

Cor Total

268.71

16

Note:**p<0.01,*p<0.05

Comment 9:

In vitro drug release: Describe the HPLC method, specifying the column type, instrumentation, and operating conditions, as well as the calibration curve, detection wavelength, and the concentration range evaluated during validation.

Our reply:

Thank you for pointing out this. We have added this section in the manuscript.

HPLC (Agilent, USA) and a Diamonsil 5 µm C18 column (Dikma, China) was used for the determination and quantification of AB4. The mobile phase combined two eluents (A:B) in a 30/70 ratio at a flow rate of 1 mL/min, with detection at 210 nm. Eluent A: acetonitrile, eluent B: water. Injection volume: 10μL. The run time was 15 min and the retention time of AB4 was approximately 12.5 min. A linear correlation was acquired between peak area and concentration. The linear equation was y=2947.1x+23660 (R2=0.9993), where x is the concentration and y is the peak area. The assay was linear in the concentration of 10~2000 µg/mL.

Comment 10:

Pharmacokinetic study: How were the blood samples analyzed? Please describe the analytical method employed.

Our reply:

Thank you for your valuable suggestions. We have added this section in the manuscript.

After the plasma sample was thawed at room temperature, 50 µL of plasma, 400 µL of methanol and 50 µL of internal standard solution (1 µg/mL Ginsenoside Rg1) were accurately weighed in a centrifuge tube and vortexed for 3 min to precipitate the proteins. After centrifugation at 13,000 rpm for 10 min at 4 ℃, the supernatant was filtered through a 0.22 µm membrane, and the plasma drug content was determined according to the above UPLC-MS/MS method. The obtained AB4 plasma concentration results were processed according to the DAS 3.0 program non- compartmental model. (Beijing JiDaoChengran Technology Co., Ltd., Beijing, China).

The Shimadzu UPLC system (Shimadzu, Kyoto, Japan) and AB SCIENX 4500 mass spectrometer (Shimadzu, Kyoto, Japan) were used to quantitatively analyze AB4 in rat biological samples. The analyses were conducted on AB SCIEX QTRAP 4500 triple quadrupole UPLC-MS/MS system (USA) operated in electrospray ionization (ESI) (−) mode. The optimized chromatographic separation of AB4 was performed by using Ul-timate XB-C18 column (50 mm×2.1 mm, 1.8 µm particle size). The mobile phase was 0.1 % formic acid aqueous solution (A) and acetonitrile (B), and the gradient elution proce-dure was 0-1.2 min, 10 % B; 0.1-0.5 min, 10-40 % B; 0.5-2.0 min, 40–95 % B; 2.0-3.2 min, 95 % B; 3.2-4.0 min, 95-10 % B; 4.0-5.0 min, 10 % B, with the flow rate maintained at 0.4 mL/min. The injection volume was set at 3 µL and the column oven was maintained at 35 ℃. The most abundant fragment ions in multiple reaction monitoring (MRM) were adopted, and m/z 1219.5→749.5 for AB4 and m/z 845.4→637.4 for Ginsenoside Rg1 at a collision energy of -180 V and -200 V were monitored. Decluster potential of AB4 and Ginsenoside Rg1 was -55 eV and -30 eV, respectively. The total run time of the assay was 5.0 min. The retention time of AB4 was 2.23 min, and the retention time of the Ginsenoside Rg1 was 2.22 min. The ion spray voltage was adjusted at 4.0 kV. The common parameters were as follows: nebulizer gas pressure was 50 psi; gas temperature was 325 °C. The linear concentration range of AB4 in rat plasma was 0.01-3.2 µg/mL with a lower limit of quantification (LLOQ) of 0.01 µg/mL (R=0.9925). The mean AB4 plasma extraction recovery was 90.16±9.83 %. The intraday precision was about 5.80% at the quantitation limit 300 ng/mL, which provided sufficient sensitivity to characterize pharmacokinetics.

Comment 11:

Rectal retention test: Please be consistent and format the bibliography uniformly throughout the document (reference 33)

Our reply:

Thank you for this question. We have revised in the manuscript .

Comment 12:

Please elaborate further on the sub-sections of the Results and Discussion section, providing appropriate references to support the claims and findings. Proper citation of relevant studies is essential to strengthen the validity of the results. I strongly recommend that the authors discuss and validate most of the results and characterization data by analyzing their implications and comparing them with findings from other studies.

Our reply:

Thank you for pointing out this. We have discussed most of the results and characterization data in the manuscript, with reference to relevant studies.

Comment 13:

Optimization of the ISG formulation: Please explain the rationale behind this statement: “HPMC had no obvious effect on gelation temperature, whereas P407 exhibited the greatest influence on gelation temperature. The effect of AB4 and the absorption enhancers HP-β-CD and SC on the gelation temperature was approximately 4°C.

Our reply:

Thank you for the question. The gelation temperature is predominantly governed by P407 concentration, as it drives the thermosensitive sol-gel transition, a finding confirmed by our experimental optimization. In contrast, HPMC (0.44%) was added to modulate viscosity, which explains its negligible impact on the gelation temperature. The soluble drugs and enhancers permeation increase the gelation temperature by approximately 4°C, which aligns with theoretical expectations. Considering the rectal temperature is approximately 37°C, we set the target tempera- ture at 32°C. This adjustment ensures the formulation gels rapidly at body temperature.

 Comment 14:

“Gelation temperature, gelation time pH, viscosity and gelation strength: Do all formulations exhibit identical viscosity and gelation strength, and what are the implications of these results?

Our reply:

Thank you for pointing out this. All formulations exhibited similar viscosity at room temperature (25℃), ensuring easy injectability, while undergoing a sol-gel transition with significantly increased viscosity at body temperature (37°C). The comparable viscosity (37℃) and gel strength across formulations indicated that the incorporation of AB4, HP-β-CD, or SC did not substantially alter the mechanical strength of the gel network once the P407/P188/HPMC matrix was established.

Comment 15:

In vitro drug release: What are the implications of all formulations fitting a first‑order release model?

Our reply:

Thank you for pointing out this. All formulations conform to the first-order release model, indicating that the drug release occurs primarily through diffusion.

Comment 16:

It is recommended to define abbreviations before using them; for example, H&E.

Our reply:

Thank you for this question. We have revised in the manuscript.

Comment 17:

Please improve the clarity of the figures by increasing the font size of all labels, legends, and axis titles.

Our reply:

Thank you for this question. We have revised in the manuscript.

Comment 18:

Please italicize the terms in situ, in vivo and in vitro everywhere they appear in the manuscript, including in the title, subtitles, and reference list.

Our reply:

Thank you for this question. We have revised italicize the terms the terms in situ, in vivo and in vitro in the manuscript.

Comment 19:

Please improve the overall presentation of the manuscript: number the lines, use a uniform font size throughout the text, insert proper spacing between numbers and their units, avoid excessive bold formatting, and attend to other typographic details.

Our reply:

Thank you for this question. We have revised in the manuscript.

Comment 20:

Comments on the Quality of English Language Finally, the English is sub‑par and the manuscript lacks clarity and logical flow, which makes it difficult for readers to follow the argument and remain engaged.

Our reply:

Thank you for your comment. We have revised the manuscript for English language.

I hope the above responses and the revised manuscript can satisfy the questions and comments from the reviewers and you. Should you have any queries, please do not hesitate to contact me at (+86) 0791-87119623. Thanks for your consideration of our work. I look forward hearing from you.

Yours Sincerely,

Guo-Song Zhang

Corresponding Author:

Prof. Guo-Song Zhang, National Pharmaceutical Engineering Center of Solid Preparation in Chinese Herbal Medicine, Jiangxi University of Chinese Medicine,  Nanchang 330006, China.

Tel: (+86) 0791-87119623

E-mail: zhgs81411@aliyun.com

Round 2

Reviewer 1 Report

Comments and Suggestions for Authors

Thank you for the authors’ valuable efforts in revising the manuscript. The paper is now suitable for publication.

Author Response

Comment 1:

Thank you for the authors’ valuable efforts in revising the manuscript. The paper is now suitable for publication.

Our reply:

We greatly appreciate your guidance!

Reviewer 2 Report

Comments and Suggestions for Authors

Thank you very much for your review. However, several aspects of the manuscript still need to be addressed:

  1. Absorption enhancers: The manuscript lacks a clear explanation of the chosen absorption enhancers and the rationale behind their selection for this study. Please include a brief discussion of their mechanisms of action and why they are suitable for the intended application.

  2. Section 3.1 – Screening of absorption enhancers: This section appears to contain a copy‑paste error; instead of describing the screening protocol, it repeats the HPLC method details. Replace the current text with a concise description of the screening assay (e.g., experimental setup, concentrations tested, evaluation criteria).

  3. Figure quality: Most of the figures are low‑resolution and appear blurry. Please provide high‑resolution versions to ensure all details are clearly visible.

Author Response

Comment 1:

Absorption enhancers: The manuscript lacks a clear explanation of the chosen absorption enhancers and the rationale behind their selection for this study. Please include a brief discussion of their mechanisms of action and why they are suitable for the intended application.

Our reply:

Thank you for pointing out this. We have added this section to the manuscript.

1.Introduction:

To overcome these deficiencies, absorption enhancers are usually used to accelerate absorption. Commonly used enhancers include cyclodextrin, surfactants, metal chelators, salicylates and their derivatives, and bioadhesive polymers, et al. These compounds can improve drug absorption either by fluidizing cell membranes to enhance transcellular transport or by reorganizing tight junctions to facilitate paracellular transport [17]. This study aims to systematically compare the absorption enhancing effects of several permeation enhancers hydroxypropyl-β-cyclodextrin (HP-β-CD), sodium caprate (SC), chitosan (CS), Tween-80, water-soluble azone (azone), and arginine (Arg) through individual application. The objective is to identify the two most effective enhancers and their optimal concentrations for maximizing AB4 permeation, thereby providing an experimental basis for future development of rectal delivery formulations of AB4.

3.1.3. The screening of absorption enhancers:

AB4 is a large molecular weight compound (1,221.38 Da) classified as a BCS III drug, which likely gets absorbed via the paracellular pathway or through enhanced membrane fluidity. This study employed Caco-2 cell permeability assays and Franz diffusion cell experiments to screen suitable absorption enhancers from among HP-β-CD, SC, CS, Tween-80, azone, and Arg. Among them, HP-β-CD not only improves drug solubility but also interacts with cholesterol and phospholipids in the cell membrane, transiently and reversibly altering membrane structure and fluidity to enhance transcellular permeability [46]. SC reversibly opens tight junctions, thereby creating paracellular pathways to promote the absorption of hydrophilic drugs. This process is reversible and typically recovers within hours. Furthermore, SC has a long history of human use with food additive status and has been extensively evaluated as an intestinal permeation enhancer for the oral delivery of macromolecules [47]. CS also reversibly opens epithelial tight junctions, though its mechanism is more associated with direct interaction with junctional proteins to facilitate paracellular transport [46]. Tween-80, a non-ionic surfactant, inserts into the lipid bilayer of cell membranes, disrupting lipid organization and increasing membrane fluidity, thereby reducing resistance to transcellular drug diffusion [48]. Azone inserts into the lipid regions of cell membranes and interacts with lipid molecules, disrupting their tight packing and making the membrane "looser" and more fluid. The positively charged Arg can bind to the negatively charged head groups of membrane phospholipids through electrostatic interactions, perturbing membrane structure [49]. Results from both Caco-2 cell permeability studies and Franz diffusion cell experiments demonstrated that HP-β-CD and SC were the most effective absorption enhancers, promoting AB4 absorption by either increasing membrane fluidity or opening tight junctions between cells, which aligns with their known mechanisms of action. Based on comprehensive consideration, 2.5% HP-β-CD and 1% SC were selected as absorption enhancers for AB4-ISG.

Comment 2:

Section 3.1 - Screening of absorption enhancers: This section appears to contain a copypaste error; instead of describing the screening protocol, it repeats the HPLC method details. Replace the current text with a concise description of the screening assay (e.g., experimental setup, concentrations tested, evaluation criteria).

Our reply:

Thank you for pointing this out. We have revised the section accordingly. Since “the screening of absorption enhancers” and “the in vitro release study” used two different standard curves, the methodological description was duplicated in the original text. To clarify, we have now added a new subsection titled “3.1.1 HPLC Method”.

Comment 3:

Figure quality: Most of the figures are low‑resolution and appear blurry. Please provide high‑resolution versions to ensure all details are clearly visible.

Our reply:

Thank you for this question. We have revised in the manuscript.